# Bioinformatics leading to conveniently accessible, helix enforcing, bicyclic ASX motif mimics (BAMMs)

Tianxiong Mi[1], Duyen Nguyen[1], Zhe Gao [1] & Kevin Burgess [1] ✉

Helix mimicry provides probes to perturb protein-protein interactions (PPIs). Helical conformations can be stabilized by joining side chains of non-terminal residues (*stapling*) or via *capping* fragments. *Nature* exclusively uses capping, but *synthetic* helical mimics are heavily biased towards stapling. This study comprises: (i) creation of a searchable database of unique helical *N*-caps (ASX motifs, a protein structural motif with two intramolecular hydrogen-bonds between aspartic acid/asparagine and following residues); (ii) testing trends observed in this database using linear peptides comprising only canonical L-amino acids; and, (iii) novel synthetic *N*-caps for helical interface mimicry. Here we show many natural ASX motifs comprise *hydrophobic triangles*, validate their effect in linear peptides, and further develop a biomimetic of them, *B*icyclic *A*SX *M*otif *M*imics (BAMMs). BAMMs are powerful helix inducing motifs. They are synthetically accessible, and potentially useful to a broad section of the community studying disruption of PPIs using secondary structure mimics.

Protein ligands frequently present interface α-helices in protein-protein interactions (PPIs)[1-3]. Interface helix *mimicry* is one of the most generally applicable ways to disrupt PPIs. Helix mimicry via stapling[4,5] features one or two peripheral fragments to produce monocyclic rings. However, most stapled peptides have exocyclic *N*- and *C*-terminal peptide regions vulnerable to conformational fraying, and proteolytic trimming by exopeptidases. Further, compositions of staples varies widely, and this variance can impact populations of true α-helical conformations induced. Staples have been studied more extensively than the alternative: synthetic caps[5,6].

Capping motifs are different to staples. They kink terminal backbones out of helicity to satisfy overhanging *H*-bond donors or acceptors. *N*-caps may flex but cannot fray, hence they protect *N*-termini from proteolytic degradation. Those properties provide advantages, but the current methodologies for installing *N*-caps have limitations.

Current routes to *N*-cap systems are inconvenient and/or produce relatively flexible systems. Early efforts to obtain helical *N*-caps involve laborious multi-step syntheses, and offer no clear advantage over the state-of-the-art[7-9]. Two unnatural *N*-capping strategies are prevalent in the contemporary literature: *H*-bond surrogates (Fig. 1a), and peptidic straps (Fig. 1b). Hydrogen bond surrogates feature synthetic fragments to replace an *Ncap*(*CO*) to *N4*(*NH*) *H*-bond ("m*Ncap*,m*N4*" using the *H*-bond nomenclature introduced in Fig. 1c). *Ncap* refers to the first *N*-terminal residue to deviate from regular helical φ,ψ values (−60 ± 15,−40 ± 15° for α-forms). *H*-Bond surrogates were originally produced using imines[10], but Arora's lab adapted that approach using alkene metathesis[11-13]. Alkene metathesis reactions, however, are not robust in peptide syntheses. Peptidic straps by Li and co-workers[14-16] are based on amides connecting the $C_{\beta}HCO_2H$ carbonyl of D-*iso*Asp (*iso*-aspartic acid) to side chain *NH* of L-Dap (diaminopropionate) residues at *N3*. Both these synthetic *N*-cap motifs comprise one monocyclic ring, hence allow significant conformational flexibility.

A prevalent *N*-cap in *Nature* is the ASX motif (Fig. 1c)[17,18]. ASX motifs are characterized by two *H*-bond contacts: one between *i* side chain *CO* of Asp or Asn and a main chain *NH* of *i* + 2 or *i* + 3, and another between the Asp or Asn main chain carbonyl to the *i* + 3 or *i* + 4 main chain *NH*. ASX motifs at helical *N*-termini can be described using conventional helical residue numbering: *Ncap* side chain *CO* to the *N2* or *N3* main chain *NH* (s*Ncap*,m*N2* or s*Ncap*,m*N3*), and from the *Ncap* main chain *CO* to the *N3* or *N4* main chain *NH* (m*Ncap*,m*N3* or

[1]Department of Chemistry, Texas A & M University, College Station, TX 77842, USA. ✉e-mail: burgess@chem.tamu.edu

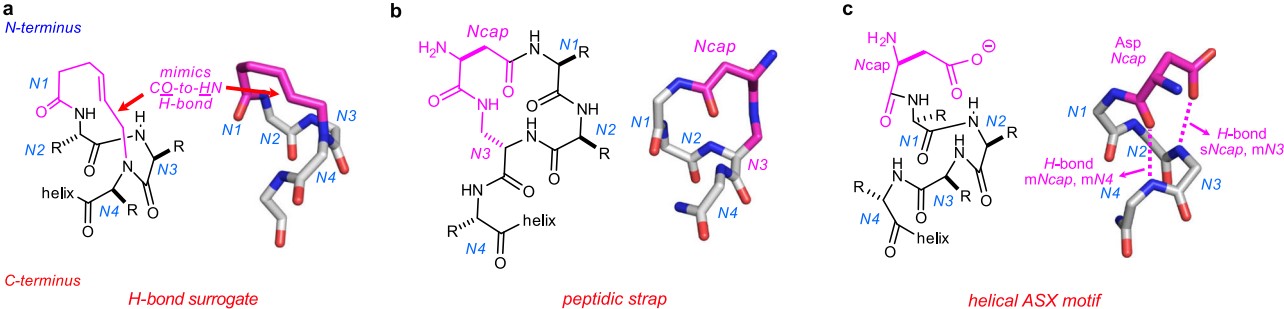

**Fig. 1 | Synthetic *N*-caps and natural ASX motifs. a** Hydrogen bond surrogate (PDB 4MZL) where an alkene mimics m*Ncap*,m*N4*. **b** Peptidic strap *N*-cap mimic (5GS4) where D-*iso*-Asp at *Ncap*, is amide-linked to L-Dap at *N3*. **c** An illustrative, natural ASX motif (1A2X) showing the two key *H*-bonds.

m*Ncap*,m*N4*). "*Ncap*" refers to the last residue with non-helical φ,ψ dihedral angles before a helix beginning, but "*N*-cap" represents capping motifs at helical *N*-termini. ASX motifs should not be confused with ASX turns. ASX turns refer to secondary structures featuring a single *H*-bond interaction in which *i* side chain *CO* of Asp or Asn *H*-bonds a main chain *NH* of *i* + 2.

A lesser known feature of ASX motifs is they are often reinforced by a hydrophobic patch between *N'* and *N4* side chains. In this work we report patches augmented to form *hydrophobic triangles* by packing side chain fragments at *N'*,*N4*, and *N3*. We also report the most accessible, rigid, and generally applicable *N*-cap motifs reported to date; they are based on mimicry of hydrophobic triangles in ASX motifs.

## Results

### Bioinformatics of ASX motifs

The existing searchable database to access crystallographically characterized ASX motifs (PDBeMotif)[19] has limited capacity to probe structural details like atom-atom distances, dihedral angles, and sequence ranges. Consequently, we devised an alternative search routine in Fig. 2a with enhanced features. Briefly, the whole PDB was downloaded and filtered based on <95% primary sequence similarities (using a published algorithm)[20,21] to leave only unique protein chains. Two criteria were used to search all ASX motifs: an Asp or Asn, AND characteristic *H*-bond pattern for ASX motifs (helical and non-helical, Supplementary Fig 1). This data set was processed to isolate ASX motifs at helical *N*-caps using DSSP (Define Secondary Structure of Proteins[22]; Supplementary Figs 2, 3). *H*-Bonds and hydrophobic contacts were detected based on distance and angle requirements (Supplementary Figs. 1, 7). We called our routine for this whole process *ASX/ST Search*.

ST motifs are the other prevalent natural *N*-caps with similar patterns to ASX motifs but use Ser/Thr as capping residues. *ASX/ST Search* can search for both, but only the ASX set is considered here. Datasets analyzed in this paper are uploaded as SI materials. Basic statistical analyses including residue abundances of different datasets are in Supplementary Figs. 5–6.

Capping boxes[23] provide an additional s*N3*,m*Ncap H*-bond (Fig. 2b), and patches where two hydrophobic side chains interacts across the first helical turn (Fig. 2c), were known to reinforce helical ASX motifs[17,18] prior to this work. *N*-Terminal hydrophobic patches[23,24] were frequently observed between *N'* and *N4*, while previous reports on *N'* and *N3* patches were rare. Consequently, a script was designed to probe potential side chain hydrophobic contacts between residue pairs (*N'*, *N3*) and (*N'*, *N4*) in a subset of ASX motifs where *N'* and *N4* were both hydrophobic amino acids; the subset includes 12,175 motifs (36% of total). Contacts between (*N3*, *N4*) were not considered because these residues were naturally adjacent. The script measured nearest carbon distances between side chains of residue pairs, with a 4.5 Å distance cut-off; side chains within this were considered for potential hydrophobic contacts. Outputs were recorded in 1D arrays, [(*N'*,*N4*), (*N'*,*N3*)] where '1' would be assigned if potential interactions could occur.

Results from above are shown in Venn diagram Fig. 2d. Of the 12,175 ASX motifs in the key data set, 71% could form at least one hydrophobic contact, and 29% could simultaneously have (*N'*,*N4*) and (*N'*,*N3*) contacts. Supplementary Fig 7b gives percentages of the four categories from [0,0] to [1,1]. The previously underappreciated hydrophobic interaction (*N'*,*N3*), occurred alone only rarely, but frequently with the (*N'*,*N4*) contact: isolated (*N'*,*N3*) contacts and co-existing ones were, 1139 and 3492, respectively. We call motifs with both (*N'*,*N4*) and (*N'*,*N3*) contacts triangles because ASX motifs are stabilized by packing three hydrophobic fragments into triangular shapes. A hydrophobic triangle from PDB 1Q5V is shown in Fig. 2e where the residues are all leucine. All hydrophobic triangles necessarily include hydrophobic patches, but the reverse relationship does not apply.

Preferred amino acids were determined for positions at *N'*, *N4* and *N3*. Figure 2f illustrates most common *N'* and *N4* combinations in hydrophobic triangles featuring Leu at either position, vaguely reminiscent of interdigitated Leu-zippers[25]. Amino acids with shorter side chains are less common, probably because these do not contact each other well across helical turns, and "extended" aromatic residues like Phe and Trp are not incorporated frequently. These observations may be useful for including hydrophobic triangles into helix-inducing ASX motifs in peptides designed to be helical.

Figure 2g shows *N3* residue abundances in hydrophobic triangles. Initially it surprised us to find Glu, Gln and Asp abundant at *N3*, but later we realized methylene groups of these residues can contribute to hydrophobic packing, even though their termini are carboxylates or amides. Nearly three quarters (780 motifs, 73%) of ASX motifs featuring hydrophobic triangles with Glu and Gln at *N3* (1062 motifs) also formed capping boxes. Capping boxes enhance ASX motif stabilities via s*N3*,m*Ncap H*-bonds which draw *N*-terminal loops (⋯*N'*-*Ncap*) and helical fragments (*N3*-*N4*⋯) closer and better oriented for hydrophobic packing. Figure 2h and i shows *N'*,*N3*, and, *N'*,*N4* Cα - Cα distances in typical triangular cores are significantly shorter when there is a capping box than when there is not; this allows the *N3* methylene groups to pack with nearby hydrophobic side chains. Therefore, capping boxes and hydrophobic triangles can mutually stabilize each other in ASX motifs (Fig. 2j).

### ASX motifs, hydrophobic patches and triangles in model linear peptides

Linear peptides were prepared to study impacts of the featured structural motifs on helicities. Systematic changes were made at *N'*, *Ncap*, *N3* and *N4* in 17-mers for incremental addition of an ASX motif, a hydrophobic patch, and a hydrophobic triangle (Fig. 3a: AAKA {**AAA**A**KA**AAAKAAAAKAW} was the parent peptide without any of these features). Four-letter labels like AAKA (for **AA**XX**KA**) represent the *N'*, *Ncap*, *N3*, and *N4* residues; other parts of the sequence were not varied, hence are not in the labels. We first replaced A at *Ncap* with D to potentially add an ASX motif, then K at *N3* was mutated to A to avoid

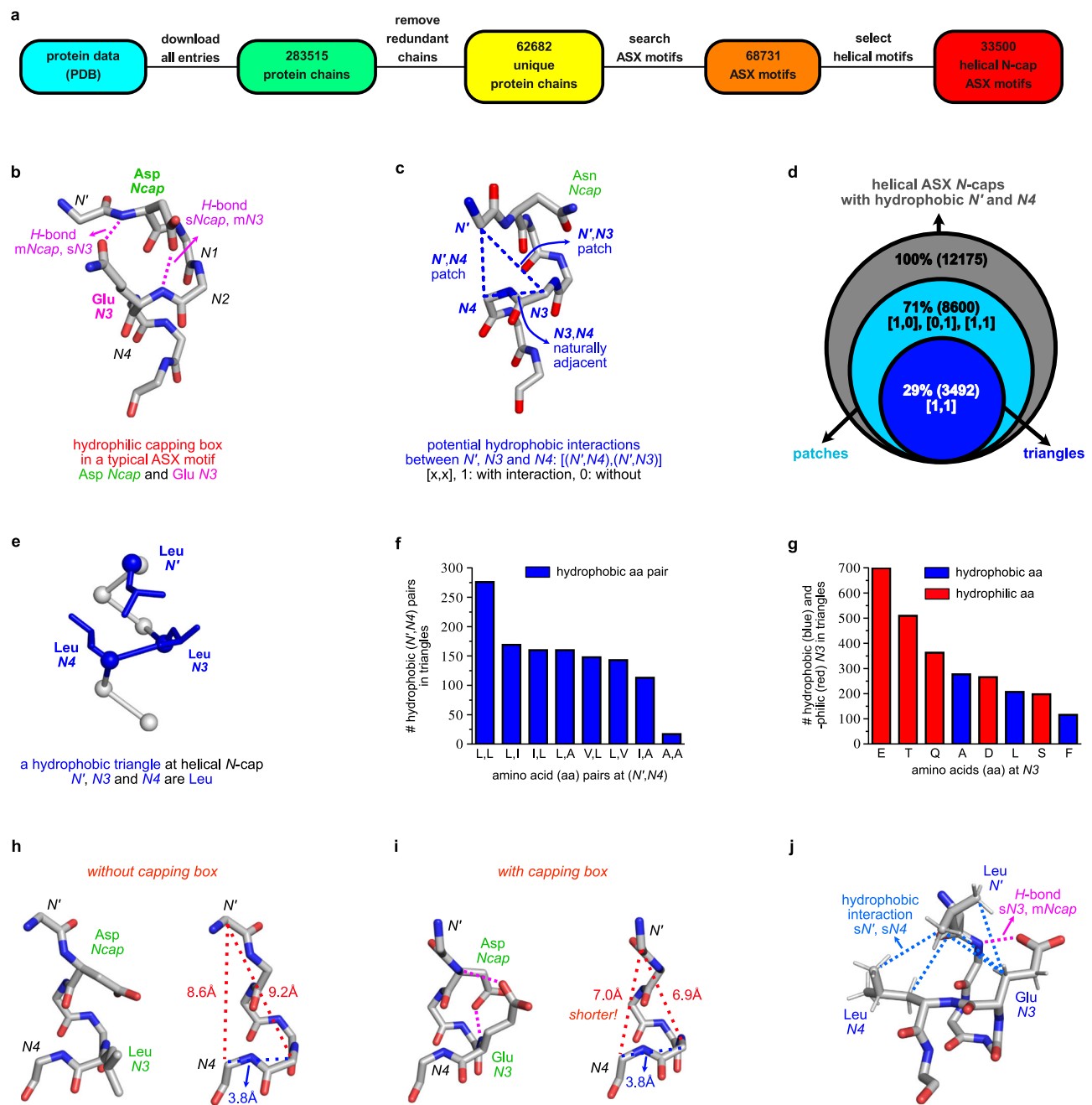

**Fig. 2 | Discovery and analyses of hydrophobic triangles. a** Search for non-redundant ASX motifs from the protein databank (PDB). **b** Structure of a typical capping box (s*N3*,m*Ncap* H-bond) incorporating an Asp *Ncap* and Glu *N3*. **c** Potential cross-turn hydrophobic interactions between side chains of (*N',N4*), and/or (*N',N3*). A 1D array, [(*N',N4*), (*N',N3*)], was used to describe hydrophobic contacts. **d** Numbers, percentages and relationships of hydrophobic patches and triangles in the key data set. **e** Illustrative hydrophobic triangle at helical *N*-cap from PDB 1Q5V.

**f** Abundance of (*N',N4*) residue pairs in triangles. (Leu,Leu) was favored at (*N',N4*: aa is amino acid, represented by their one-letter codes). **g** Abundance of amino acids at *N3* in hydrophobic triangles. Hydrophobic methylene groups can be donated by conventionally *hydrophilic* residues to form triangles. **h** *N' − N3* and *N' - N4* distances in the absence of a capping box; and, **i** with one. **j** *N3* Glu can simultaneously facilitate hydrophobic triangles and capping boxes. Source data are provided as a Source Data file.

extra H-bond contacts with D, giving ADAA. Then abundant residue pairs of hydrophobic amino acids in Fig. 2f were installed at (*N',N4*) to form hydrophobic patches; LDAL represents the most abundant pairing, (L,L). Finally, A at *N3* was replaced by the set of abundant amino acids to build a hydrophobic triangle (Fig. 2g); this gave the most helical system LDLL. Figure 3b shows CD spectra of AAKA, ADAA, LDAL, and LDLL in PBS. CD of the other combinations is in Supplementary Fig 17, and a summary of sequences and calculated helicities is in Supplementary Table 10.

Gradually increasing $\theta_{222}$ and ($\theta_{222}/\theta_{208}$) ratios from AAKA to LDLL correspond to enhanced helicity on adding an ASX motif, a patch, then a triangle to the parent sequence (Fig. 3b). LDLL, which has an ASX motif and a three-Leu hydrophobic triangle, has 93% relative helicity in PBS buffer, as indicated by a method widely used to estimate peptide helicity[26,27]. Systematic Ala substitutions at these four key positions resulted in significantly diminished helicities (Fig. 3c), corresponding to disruption of either the ASX motif or hydrophobic triangle.

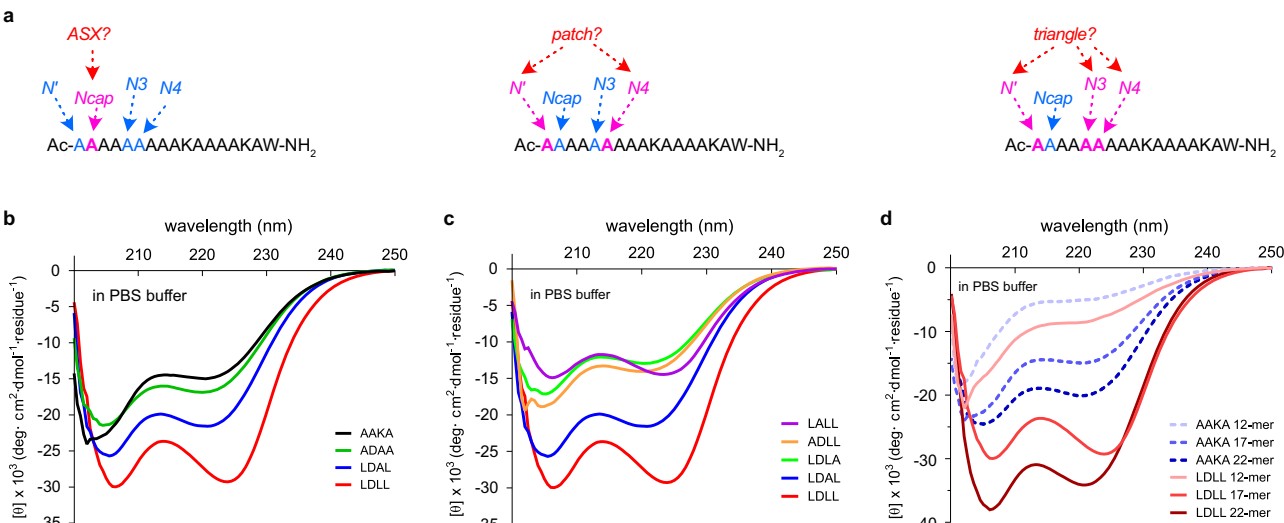

**Fig. 3 | Model Linear Peptides. a** Systematic changes at *N'*, *Ncap*, *N3* and *N4* to incrementally incorporate ASX motifs, hydrophobic patches and triangles. CD spectra of (**b**) AAKA, ADAA, LDAL and LDLL; (**c**) LALL, ADLL, LDLA, LDAL and LDLL; and, (**d**) 12-, 17- and 22-mers AAKA and LDLL. Throughout peptides were in PBS buffer at 25 °C. Source data are provided as a Source Data file.

Besides 17-mers AAKA and LDLL, 12- and 22-mer derivatives were prepared and studied (sequences in Supplementary Table 10). However, maximal helicity enhancements occurred in 17-mers where relative percent helicities increased by 45% (from 48 to 93; Fig. 3d) upon addition of a hydrophobic triangle. Less improvement (11%) was observed for the shorter 12-mers, suggesting the hydrophobic triangle in LDLL induces helicity most effectively in medium-length, partially helical peptides relative to more disordered shorter ones. This diminished helix-induction by hydrophobic triangles may arise because loose Van der Waals interactions are vulnerable to alternative conformations to maximize solvation energies and electrostatic interactions in disordered peptides. Overall, these observations inspired us to design a rigid covalent mimic of hydrophobic triangles, as described in the following section.

## Bicyclic ASX motif mimics (BAMMs)

Frequent occurrence of hydrophobic triangles in protein ASX motifs, and their unexpectedly high impact on model peptides, led us to hypothesize replacement of the hydrophobic *N'*, *N3*, and *N4* side chains by a single hydrophobic fragment covalently linked to all three residues would rigidly stabilize the helical *N*-cap. We saw a way to achieve this using CLIPs (Chemical LInkage of Peptides onto Scaffolds) chemistry[28–31]. Chemoselective CLIPS reactions of electrophiles with Cys and Cys-derivative thiols is emerging as the peptidic equivalent of click reactions[32] in organic methodology.

CLIPs can feature electrophiles with two benzylic bromomethyl groups per benzene ring, or three to form bicyclic systems[33]. Among the latter, 1,3,5-tris(bromomethyl)benzene (TBMB) is privileged insofar as all three electrophilic sites are equivalent so reactions with three inequivalent Cys residues can produce only one isomer. We envisaged a peptide with three Cys residues spaced *i*−1, *i* + 3, *i* + 4 relative to Asp (*i*) in a non-helical ASX motif would assemble into a constrained, bicyclic, helical ASX motif mimics (BAMMs) on CLIPs with TBMB (Fig. 4a) via alkylation of the three Cys residues giving a TMB {1,3,5-tris(methylene)benzene} fragment as shown.

Molecular dynamics (MD) simulations were used to test the hypothesis outlined above, and probe Cys stereochemistry dependencies. A 13-mer based on an ASX motif (hydrophobic triangle VFW: Ac-**V**DAA**FW**AAAAAAA-NH₂) was subjected to MD (298 K, 1 bar, explicit water, using Desmond from *Schrödinger*) for 50 ns. Helix persistence times were quantitatively monitored for this control linear sequence using a custom procedure designed by us (see method section); that simulation is represented by the first (pink) bar in Fig. 4b. Same procedure was repeated on eight 13-mers where the natural hydrophobic triangles were replaced by synthetic bicyclic caps with different L-Cys and D-Cys permutations. Peptide CCC with three L-Cys residues best stabilized helicity relative to the linear control VFW and all other Cys stereochemistry permutations. Specifically, CCC unwound in 21.3 ns (other data normalized to this, Fig. 4b), cCC (lower case denotes D-amino acid) was second slowest, while peptides with D-Cys at *N3* and/or *N4* unwound even faster. We expected helical conformations to tolerate D-Cys at *N'* more than *N3* and/or *N4* because *N'* is outside the helix whereas the *N3* and *N4* are within. Based on these observations, LLL-Cys combinations were used henceforth in this study.

## Syntheses and CD studies

Bicyclo 12-mer, W**C**DAA**CC**AAAKA-NH₂, was prepared to measure its helicity by circular dichroism (CD) spectroscopy. Bold italics ***C*** indicates L-Cys alkylated with TBMB to coincide with hydrophobic triangle at *N'*, *N3* and *N4*. Uncertainty regarding CLIPs efficiencies to form the bicyclic *N*-caps was dispelled since crude product HPLC analyses showed only one significant peak (Supplementary Fig 14).

Comparisons were then made for the bicyclo 12-mer relative to two linear controls: AAKA 12-mer, a negative without structural motifs, the linear LDLL 12-mer which has an ASX motif and hydrophobic triangle but no rings, and two monocyclic controls (Fig. 4c). The two 12-mer monocyclic controls have ring sizes corresponding to the BAMM bicycle, *ie* a wide *N'* - *N4* linkage and a narrower one which connects *N'* - *N3*. BAMMs contain three rings, but the third conceivable control is irrelevant because it would connect adjacent *N3* and *N4* residues which could not stabilize a helical turn.

Key CD data for peptides in 10%TFE/PBS are in Fig. 4d, while Fig. 4e and Supplementary Table 11 compare relative percent helicities and molar ellipticity ratios ($\theta_{222}/\theta_{208}$) in 10%TFE/PBS and in PBS. The bicyclic *N*-cap proved to be significantly more helical than all the controls, in both solvents. Conspicuously less helix induction by both monocyclic controls is intriguing because this shows effects of the bicyclic system. Moreover, the narrow *N'* - *N3* monocyclic system represents an *i* - *i*+4 staple, so the bicyclic system is more helix inducing than a closely related stapled system. CD experiments measured in 20% TFE/PBS show greater ellipticity at 222 nm and 222/208 ellipticity ratios for bicyclo 12-mer (Supplementary Fig 19) than typical

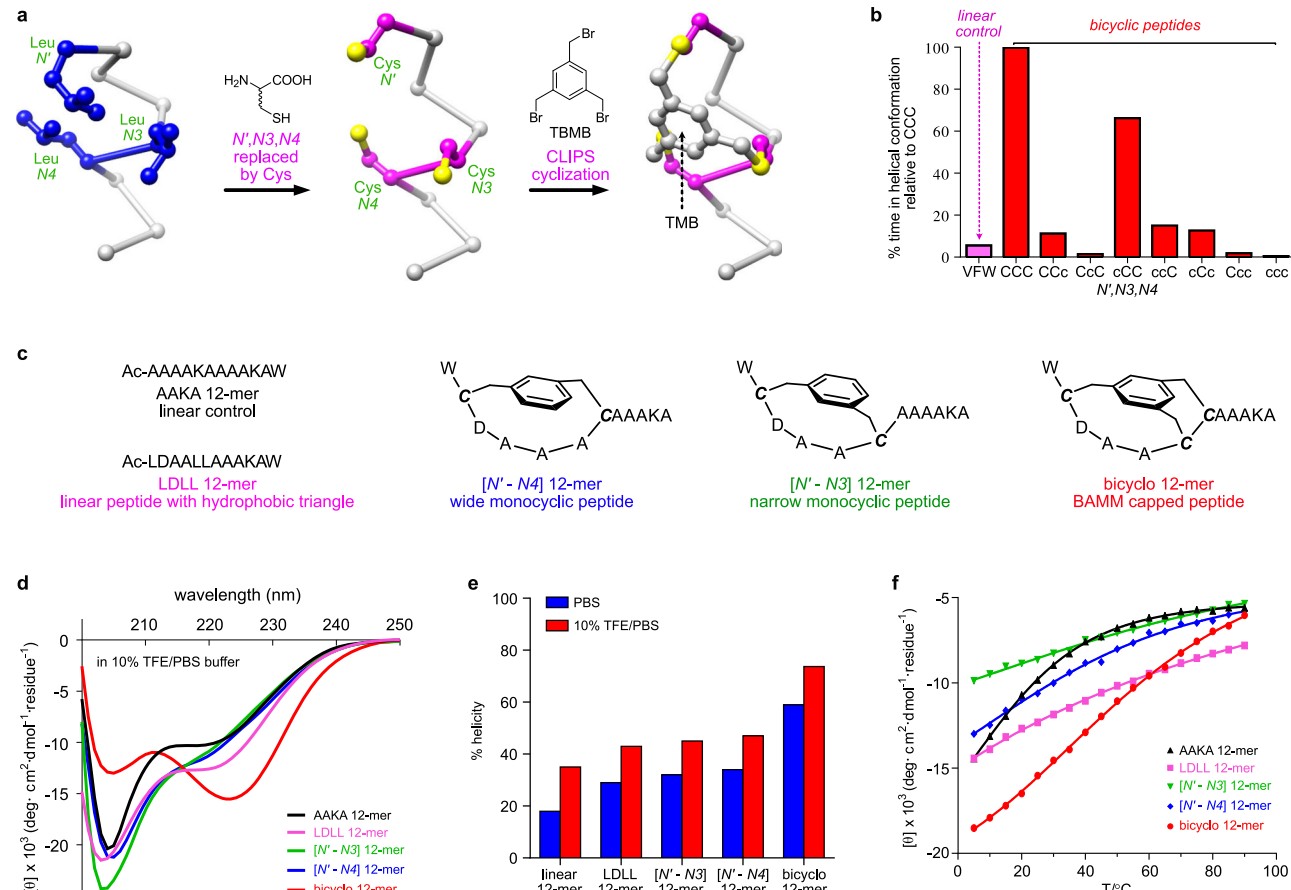

**Fig. 4 | Bicyclic ASX motif mimics (BAMMs). a** Relationship between hydrophobic triangles and BAMMs. TBMB is 1,3,5-tris(bromomethyl)benzene, and CLIPS for Chemical LInkage of Peptides onto Scaffolds. **b** Quantitated helical residence times in MD for a 13-mer linear control VFW and BAMMs with all possible Cys-stereochemical combinations at 27 °C over 50 ns. CCC, residence time 21 ns, was set at 100% for normalization. "C" is L-Cys, and "c" D-Cys. **c** Control peptides without (linear) or with only one (monocyclic) ring constraint compared with the bicyclic 12-mer. **d** CD data for two linear and two monocyclic controls and the bicyclic 12-mer at 25 °C in 10% TFE/PBS. **e** Calculated relative %helicities at 25 °C in PBS (blue) and 10% TFE/PBS (red). **f** $\theta_{222}$ for the featured peptides in 10% TFE/PBS from 5 to 90 °C. Source data are provided as a Source Data file.

α-helices, as expected from "the curvature effect" for Ala-rich helical peptides[34]. Throughout, CD studies were also performed at different concentrations to probe for aggregation of BAMM bicyclic peptides (Supplementary Fig 22), and no evidence for it was observed.

Variable temperature CD (VT-CD) experiments (Fig. 4f and Supplementary Fig 23) showed bicyclo 12-mer to be significantly more helical at most temperatures. Thermodynamic data might be extracted via the accepted Van't Hoff-like strategy[35], but this would be misleading because, unlike bicyclo 12-mer, the four control peptides were far from helical states even at low temperature in 10% TFE/PBS; these controls had much weaker helix-inducing effects than bicyclo 12-mer.

Two other bicyclic peptides were prepared to test changing the Asp *Ncap* (Supplementary Fig 20). For bicyclo-D/A 12-mer (WC**A**AAC-CAAAKA), substitution by Ala reduced helicity in PBS by 49%, where that change negates the ASX *H*-bonding pattern. In contrast, Asp to Asn substitutions had less predictable consequences because both residues form similar *H*-bonding patterns in ASX motifs. The Asp containing system was more helical perhaps because the {negatively charged} Asp carboxylates are better *H*-bond acceptors than Asn amide side chains.

## NMR studies
NMR experiments (20% TFE-d₃/5% D₂O/75% H₂O) were performed on LDLL 12-mer and bicyclo 12-mer. Enhanced Monte Carlo Multiple Minimum (MCMM) calculations[36] using NMR constraints gave 23 and

50 conformations within 3 kcal•mol⁻¹ of the minima, respectively (Fig. 5a, d, Supplementary Figs. 32, 39). Both show minimal divergence, indicating important {populated} conformers are essentially homogenous and exhibit triangulated hydrophobic packing; LDLL 12-mer via Van der Waals interactions, and bicyclo 12-mer via convalent linking (Fig. 5b, c, e).

Two abnormally high upfield chemical shifts observed for *N'* Cys *CαH* and *Ncap* Asp *NH* are consistent with the NMR-derived low-energy conformers: both hydrogens are in the TMB aromatic field (Fig. 5f and Supplementary Fig 40). A larger upfield shift for *N'* Cys *CαH* than for *Ncap* Asp *NH* correlates with the former hydrogen being closest to TMB.

Ramachandran plots for preferred conformers are Fig. 5g, h. Dihedrals of both *Ncap* residues (Asp, blue dots) deviate from typical helical φ,ψ values as expected for '*Ncap*' residues. Middle residues (red dots) are mainly within α-helix region, but dihedrals are more concentrated around ideal helical φ,ψ values in bicyclo 12-mer, indicating more compact helicity. Dihedrals φ,ψ of the *C*-terminal A⁹ and W¹⁰ are outside the helical region for the linear peptide, but those *C*-terminal residues for the bicyclo 12-mer (K⁸ and A⁹) are less distinguishable from the mid-helix, indicating helix induction in the synthetic mimic is stronger.

Bicyclo 12-mer's NOE constraints are consistent with its amide coupling constants (Supplementary Table 6) where consecutive coupling <6 Hz indicates helicity after *Ncap*, while larger values for *N'* and *Ncap*, suggest *N*cap is the helix breaker. Thus the BAMM *N*-cap

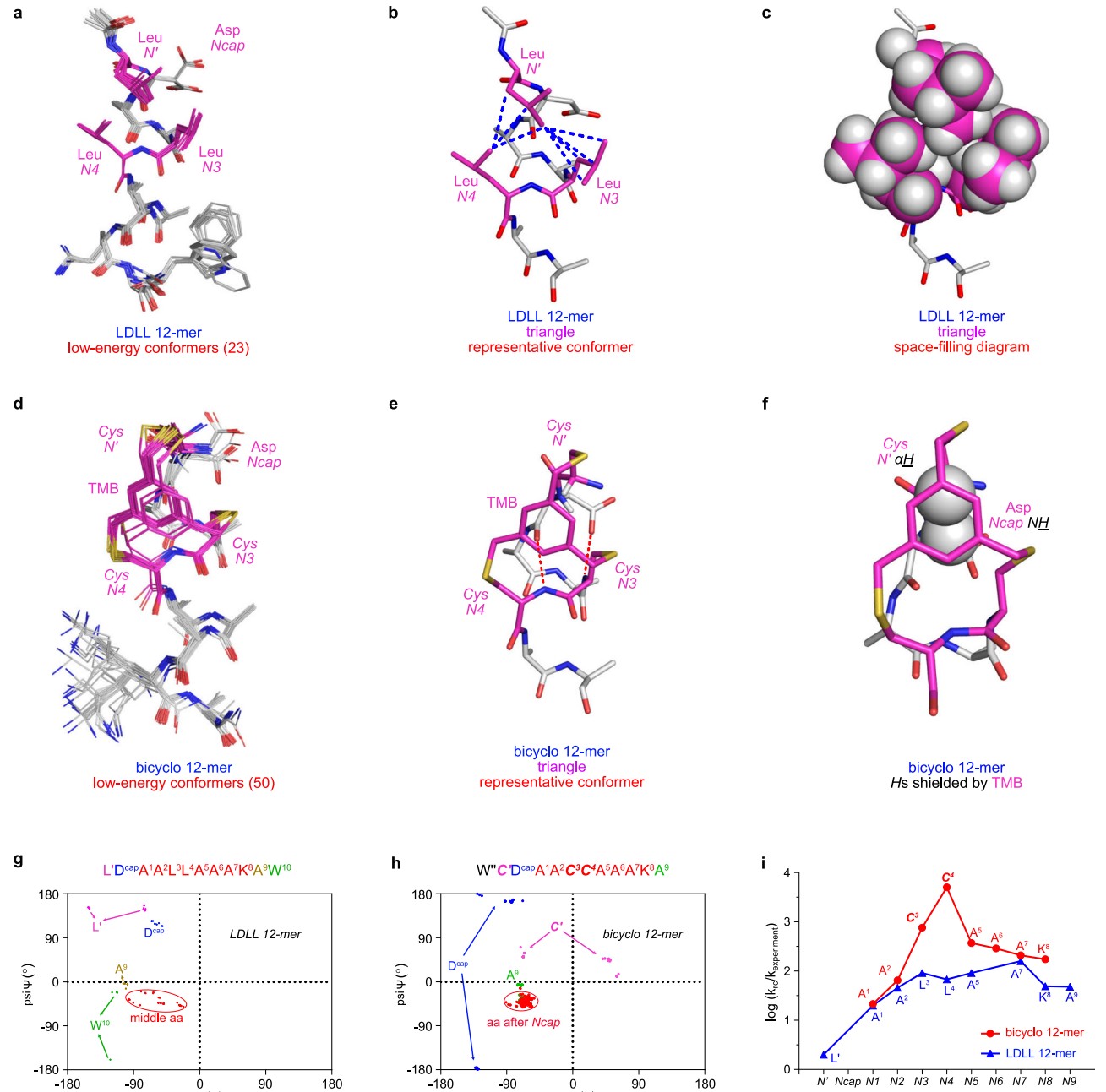

**Fig. 5 | Solution structures of LDLL and bicyclo 12-mers from conformational sampling with NMR constraints. a** Comprehensive ensemble of structures within 3 kcal•mol⁻¹ of the minimum energy conformer located for the LDLL 12-mer. **b, c** Stick and space-filling diagrams of representative conformers of LDLL 12-mer highlighting three Leu side chains comprising the hydrophobic triangle. Blue dotted lines for hydrophobic interactions. **d, e** Corresponding diagrams for the bicyclo 12-mer where the TMB fragment mimics the hydrophobic triangle. Red dotted lines

highlighting *H*-bonds in ASX motifs. **f** *N′* Cys C*α*H and *Ncap* Asp *NH* of the bicyclo 12-mer are both shielded by TMB aromaticity, with C*α* being closest. Ramachandran plots of the low energy conformers for linear LDLL (**g**), and bicyclo (**h**) 12-mer. **i** Protection factors for the bicyclo 12-mer around the capping region are significantly higher than for the linear LDLL 12-mer consistent with reduced H/D exchange rates corresponding to bicyclo *N*-cap rigidity. Source data are provided as a Source Data file.

facilitates transition from non-helical (···*N′-Ncap*) to helical residues (*N1-N2*···), just as in natural ASX motifs.

Figure 5i shows log *H/D* exchange rate ratios for random coil over test structures[37]. *N4* and *N3 NHs* in the LDLL 12-mer and bicyclo 12-mer are relatively protected from the aqueous environment, because both contain ASX motifs to shield *NH* atoms of *N3* and *N4*. Protection factors for the BAMM peptide are higher, indicating the covalent bicyclic cage is more static and helix-inducing than the noncovalent hydrophobic triangle.

## BAMM-containing analogs of natural interface helices

Interface helices can interact with protein receptors in at least the three different ways represented in Fig. 6a. If the *N*-terminus of the helix is partially buried by the receptor, (i) *non-interacting residues*, *N3* and *N4* of the helix, may be mutated and a dipeptide fragment can be appended ahead of the helix to form a BAMM. Alternatively, if the native helix totally protrudes beyond the interface as in (ii) then residues are substituted to form a BAMM overlapping that region. Occasionally ligand interface helices are totally receptor encapsulated, as in (iii), then a

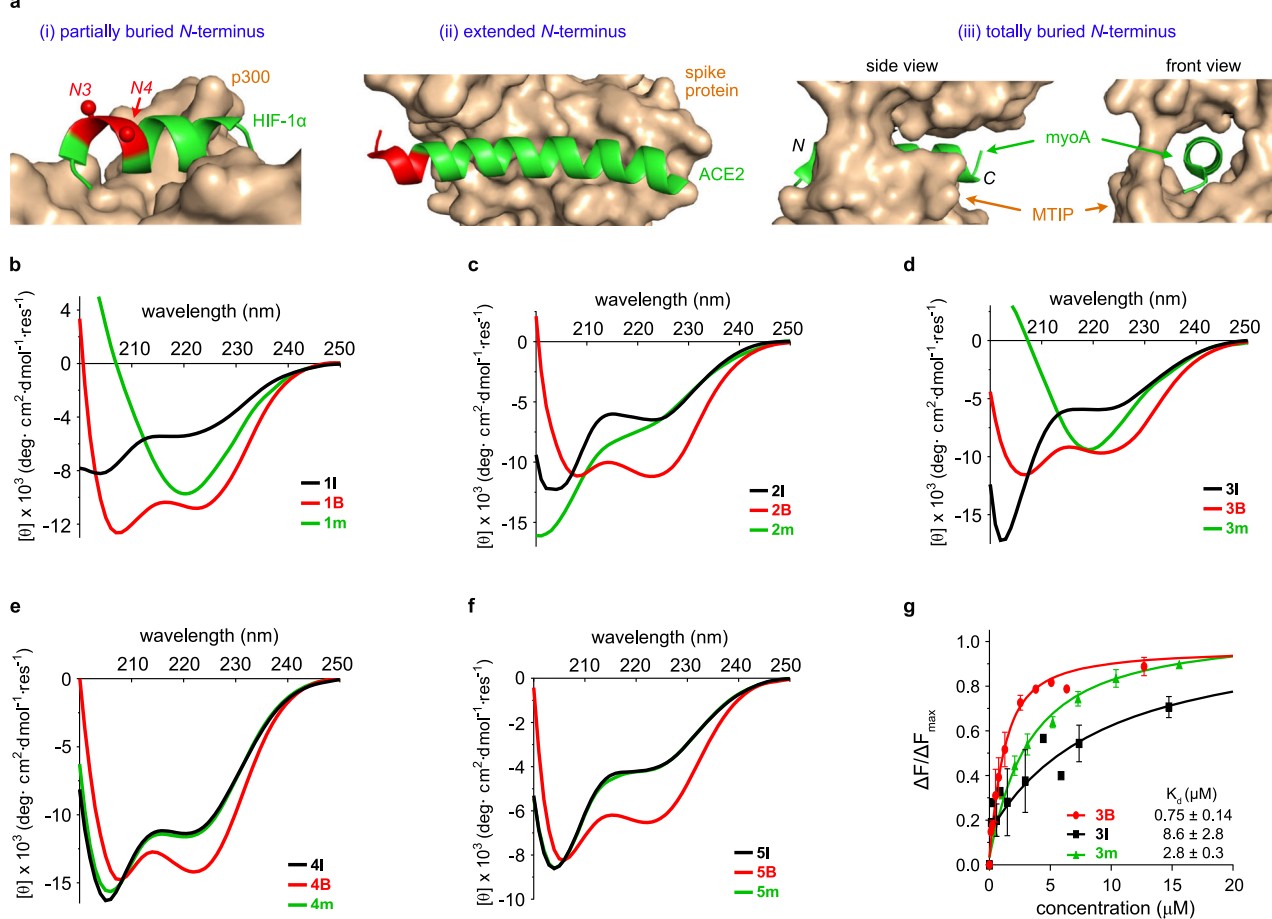

**Fig. 6 | BAMM mimics of natural interface helices. a** Three ways in which interface helices interact with their receptors (gold) corresponding to (i)–(iii) in text (PDB 1L8C, 6M0J, 4AOM). Residues which may be mutated in red. **b–f** CD spectra of **1** (interface helix from Bcl-2 homology 3 domain (BH3) in BH3•B-cell lymphoma-extra-large {Bcl-xL}), **2** (from hypoxia-inducible factor 1 alpha {HIF−1α} in HIF−1α• E1A-binding protein p300), **3** (from Cyclin-dependent kinase 2 {CDK2} in CDK2•Cyclin E), **4** (from myosin A {myoA} in myoA• MyoA tail interacting protein (MTIP)), **5** (from angiotensin-converting enzyme 2 {ACE2} in ACE2• Receptor-binding domain (RBD) of spike protein) at 25 °C in 10% TFE/PBS. **g** Relative fluorescence intensity changes as a function of **3** series peptide concentrations. ΔF for fluorescence intensity changes. $n = 2$ independent experiments, error bar = S.D. Source data are provided as a Source Data file.

BAMM-module could be added to the *N*-terminus giving a mimic longer than the native sequence.

So far this paper has described BAMMs applied to Ala-rich 12-mers. Mimicry of five naturally occurring helices comprising 10, 12, 14, 19, and 27 amino acids, **1–5**, respectively, was also studied to illustrate the situations described above[11,38–41]. There was no evidence isolated peptides corresponding to any of these would be helical in solution. Additionally, some contain Gly and/or Pro helix breakers, so peptide analogs with Gly and Pro mutated to Ala were made. Despite these helix-inducing mutations, only one peptide control, the 19-mer (**4 l**, see below for numbering), showed any evidence of helicity.

The 10, 12, and 14-mers described above correspond to partially buried helices: case (i). Our objective was to induce helicities in these sequences. BAMM mimics necessarily have two non-helical residues, so BAMM-containing analogs having two more residues in each case, *ie* 12, 14, 16-mers, were made.

A totally buried case (iii) is illustrated by the native 19-mer. A BAMM module totally outside this sequence was added to illustrate how to avoid steric hindrance with receptors which totally enclose a helix. Totally, BAMMs have six residues so a 25 residue mimic was made.

An extended helix case (ii) is illustrated by the 27-mer. For this the five *N*-terminal residues that do not contact the receptor were replaced by a six residue BAMM module, giving a 28-mer mimic. Full sequence information for all controls and BAMM-containing mimics are in Supplementary Table 12.

Two unconstrained controls were made corresponding to each BAMM mimic. The first type corresponds to interface helices with a few substitutions for helix breakers, as described above; these are **1 l, 2 l, 3 l, …..** etc. The second type corresponds to sequences of BAMM mimics except *N′*, *N3*, and *N4* were *S*-methyl Cys, thus controlling for necessary extensions and mutations from linear peptides to BAMM mimics; these are **1 m, 2 m, 3 m, …..** etc. Attempts to use free Cys rather than *S*-methyl Cys were impractical due to partial oxidation to disulfides during preparation, purification, and biophysical assays.

Supplementary Fig 21 shows CD spectra for the BAMM mimics in PBS buffer gave helical improvements over controls, but these were more significant in 10% TFE/PBS (Fig. 6b–f). Briefly, helicities of BAMM-containing mimics **B** (red) are all appreciable, whereas most linear **l** (black) and **m** (green) controls show little helical character. Interestingly, CD spectra of uncyclized linear peptides **1 m** and **3 m** implied dominance of β structures, but **1B** and **3B** significantly favored helical conformations upon cyclization to BAMM *N*-caps.

Supplementary Fig 44–53 show serum stability studies; these were monitored by analytical HPLC and LC-MS to enable detection of degradation products. BAMM containing mimics (red) prevented degradation from the *N*-termini, and stability enhancements relative to controls were greatest for shorter systems (**1–3**) where BAMM can block peptidases most effectively. Stability enhancements are less significant for longer peptides (**4 and 5**) where the BAMM unit is removed, hence less protective of the *C*-terminus with respect to

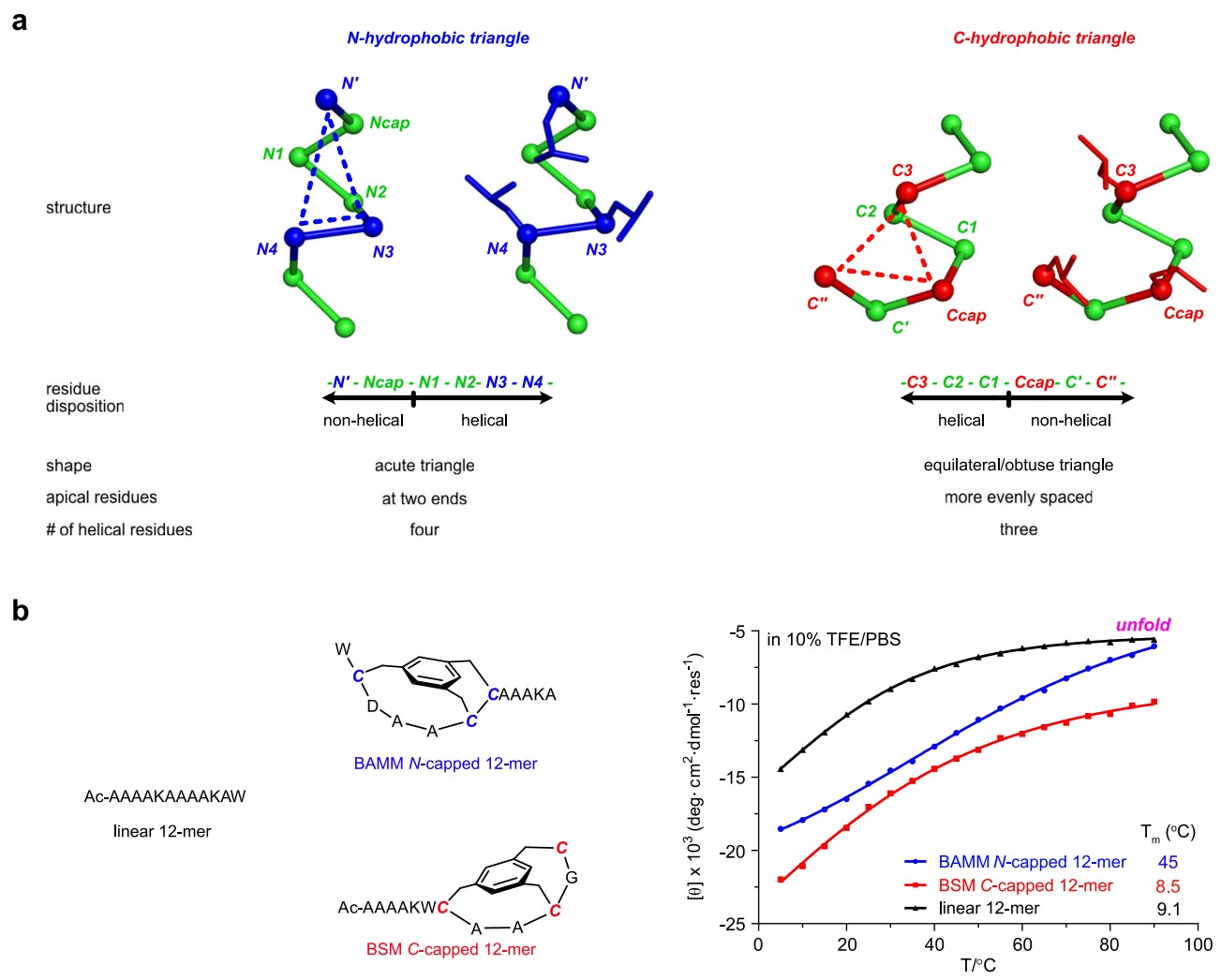

**Fig. 7 | BAMM and BSM caps have different structures and effects on formation and stabilization of helices. a** Differences between *N*- and *C*-terminal hydrophobic triangles. **b** VT-CD experiments show 222 nm ellipticity for BAMM *N*-capped, BSM *C*-capped systems (blue and red, respectively) relative to linear controls (black) from 5 to 90 °C. Source data are provided as a Source Data file.

degradation by *C*- and *endo*-peptidases. Throughout, BAMM motifs tend to be degradation end points (LC-MS); in some cases we were able to observe hydrolysis intermediates leading to them.

The peptides and mimics numbered **3** are longer than two, and shorter than the other two of the five native interface helices comprising 10, 12, 14, 19, and 27 amino acids, 1–5, so **3** was chosen for further studies. They correspond to a PPI of special interest: the helix of CDK2 interacting with cyclin E. Currently, there are no reports of helical mimics for the CDK2•cyclin E interface so we chose to look at series **3** binding cyclin E. Two binding assays were performed, the first being fluorescence quenching of cyclin E Trp224 (Supplementary Fig 54)[42,43]. That residue interacts with the CDK2 helix, so its fluorescence is particularly binding sensitive. In the event, the BAMM mimic $K_d$ (0.75 μM) was significantly better than either control (2.8 and 8.6 μM). A complementary binding assay based on fluorescence polarization (FP, Supplementary Fig. 55) was also performed. This required dye-labeled analogs, so **3lf, 3mf,** and **3Bf** were prepared which bear *N*-terminal FITC (fluorescein *iso*-thiocyanate). Binding data for these labeled peptides (Supplementary Table 8) shows the same trend as the fluorescence quenching assay. Moreover, BAMM-induced improvement of binding affinity appear to be comparable with those from *H*-bond surrogate mimics. Direct comparisons could not be made with the literature data because the protein receptors are different, but

improvements over linear controls were comparable (analysis in Supplementary Table 9)[44,45].

In conclusion, CD suggested BAMM *N*-cap can efficiently enhance α-helicity for biological sequences from 10-mer to 27-mer; serum stability assays suggested BAMM *N*-cap can prolong half-life of biological sequences, by preventing peptides from degrading from *N*-termini (intact BAMM-containing *N*-terminal fragments prevailed in degradation profiles). Helix-inducing and degradation resistance of BAMM *N*-caps decrease as peptides grow longer, as expected.

## BAMMs and BSMs compared

Recently we reported bioinformatic studies leading to the discovery of *C*-terminal hydrophobic triangles in Schellman loops[46] giving synthetic *C*-caps, Bicyclic Schellman loop Mimics (BSMs). Bioinformatic studies of *N*-terminal hydrophobic triangles in ASX motifs described here feature similar approaches but give fundamentally different findings in the following ways (Fig. 7a):

(i) the *N*-terminal hydrophobic triangles comprise four helical residues, *N1 – N4*, whereas the *C*-terminal ones have only three: *C3 – C1*;

(ii) *N*-terminal hydrophobic triangles tend towards isosceles, but at the *C*-terminus they are more closely equilateral; and,

(iii) apical residues in *C*-hydrophobic triangles are more evenly spaced allowing their side chains to pack more symmetrically.

Consequently, bioinformatic studies at *N*-termini were necessary. BAMMs could not have been designed by analogy with BSMs, because their residue dispositions, degree of incorporation into helices, and triangular shapes are different.

Further, we believe these observations account for VT CD data shown in Fig. 7b and impact a larger issue. Figure 7b shows the BSM-containing peptide is more stable at all temperatures than ones based on BAMM and the control comprising only natural amino acids; thus the rigidified *C*-capped system is more thermodynamically stable. Conversely, and perhaps counter-intuitively, fitting the VT CD data to a two state model (helical folded in equilibrium with more random unfolded)[35] reveals the BAMM-containing mimic has a significantly higher melting temperature than the BSM and linear control; the latter two were seen to be significantly lower and almost the same. Thus, BAMMs are more helix-inducing but less -stabilizing than BSMs: despite being more random at high temperature, BAMM *N*-capped 12-mer folded into helical conformations faster and much earlier than linear and BSM *C*-capped 12-mer peptide. In conclusion, BAMMs and BSMs are complementary with respect to inducing and stabilizing helical conformations.

## Discussion

Our searchable database of ASX motifs revealed existence of *N*-hydrophobic triangles (Fig. 2), and revealed hitherto unknown trends towards hydrophobic triangles. This surprised us because ASX motifs have been known for more than two decades. Experiments on linear peptides proved hydrophobic triangles had a more significant helix-inducing effect on 17-mers than on 12-mers (Fig. 3). We envisaged rigid mimic of hydrophobic triangles might overcome the unfolding tendency in short peptides and continuously induce helicity, thus BAMMs were conceived.

BAMMs are the first synthetic bicyclic helical *N*-capping motifs, and the first mimics deliberately designed on ASX motifs. BAMMs are bicyclic, and inherently more rigid than monocyclic HBS and peptidic straps. Of these three, only peptidic straps and BAMMs allow for *N*-terminal extension.

BAMM modules significantly induce α-helicity and increase serum stabilities. For CDK2•cyclin E, a BAMM-containing mimic of the cyclin E interface helix improves the binding affinity by ~8x compared with the linear, wild-type peptide.

In our view, one of the overwhelming literature biases towards stapled helix mimics over capped ones is due to synthetic accessibility. Stapling via alkene metathesis necessitates stereochemically pure pentenyl alanine or pentenyl glycine, and *H*-bond surrogates require formation of *N*-allyl amino acids; commercial availability of all these amino acid derivatives can be problematic. On the contrary, CLIPs-derived BAMMs are as accessible as any staple, and more so than most or any HBS. Further, they can be obtained without using expensive reagents or ones that leave trace heavy metal impurities, unlike caps or staples prepared via ruthenium or palladium catalysis. Even relatively low levels of trace heavy metal contaminants are unacceptable for good manufacturing practice (GMP) of clinical samples. Therefore, BAMMs are more accessible and have less disadvantages than *H*-bond surrogates and staples featuring hydrocarbon linkers.

Helix-inducing and proteolytic stabilizations effects of BAMMs are greatest at the *N*-termini because their *C*-termini are less constrained and protected (Fig. 6). Combination of BAMMs and BSMs leads to the tantalizing and now realizable possibility of using two synthetic caps on a single peptide sequence to give exceptionally helical and proteolytically stable mimics. Studies of BAMM/BSM dual-capped systems are now published from our laboratory[47]. Combinations of BAMMs and BSMs have more favorable effects than either alone, and such dual capped helical mimics potentially eclipse stapling and *N*-capping strategies for interface helix mimicry.

## Methods

### Datamining to find unique natural ASX/ST Motifs in Protein Databank (PDB)

(i) All entries from PDB were downloaded to a local computer. Till Nov. 2020, there were $1.7 \times 10^5$ entries.

(ii) $6.0 \times 10^5$ chains were found in these entries. Clustering calculation based on sequence similarity for these chains was implemented using CD-HIT web server[20,21] with an identity cutoff = 0.95. It is assumed that chains with such high sequence similarity would be nearly identical with a few mutation spots and thus should have similar conformations and only need to be considered once. At last, $6.3 \times 10^4$ unique chains were obtained for further analysis.

(iii) *N*-cap motifs were categorized into 8 classes based on their hydrogen bond patterns[17,18]. As a result, for each nonredundant chain, *N*-cap motifs were searched based on the same criteria (*d* for distance, ∠ for angle):

 a. Class 1 (C1)
 i. $N_i$ is Asp or Asn (ASX motif); Ser or Thr (ST motif)
 ii. $d(O_{side} - N_{i+2}) < 3.5\,Å$
 iii. $∠(O_{side} - H - N_{i+2}) > 140\,°$
 iv. $d(O_{side} - N_{i+3}) > 3.5\,Å$ or $∠(O_{side} - H - N_{i+3}) < 140\,°$
 v. $d(O_i - N_{i+3}) < 3.5\,Å$
 vi. $∠(O_i - H - N_{i+3}) > 140\,°$
 vii. $d(O_i - N_{i+4}) > 3.5\,Å$ or $∠(O_i - H - N_{i+4}) < 140\,°$

 b. Class 2 (C2)
 i. $N_i$ is Asp or Asn (ASX motif); Ser or Thr (ST motif)
 ii. $d(O_{side} - N_{i+2}) < 3.5\,Å$
 iii. $∠(O_{side} - H - N_{i+2}) > 140\,°$
 iv. $d(O_{side} - N_{i+3}) > 3.5\,Å$ or $∠(O_{side} - H - N_{i+3}) < 140\,°$
 v. $d(O_i - N_{i+3}) < 3.5\,Å$
 vi. $∠(O_i - H - N_{i+3}) > 140\,°$
 vii. $d(O_i - N_{i+4}) < 3.5\,Å$
 viii. $∠(O_i - H - N_{i+4}) > 140\,°$

 c. Class 2a (C2a)
 i. $N_i$ is Asp or Asn (ASX motif); Ser or Thr (ST motif)
 ii. $d(O_{side} - N_{i+2}) < 3.5\,Å$
 iii. $∠(O_{side} - H - N_{i+2}) > 140\,°$
 iv. $d(O_{side} - N_{i+3}) < 3.5\,Å$
 v. $∠(O_{side} - H - N_{i+3}) > 140\,°$
 vi. $d(O_i - N_{i+3}) < 3.5\,Å$
 vii. $∠(O_i - H - N_{i+3}) > 140\,°$
 viii. $d(O_i - N_{i+4}) > 3.5\,Å$ or $∠(O_i - H - N_{i+4}) < 140\,°$

 d. Class 3 (C3)
 i. $N_i$ is Asp or Asn (ASX motif); Ser or Thr (ST motif)
 ii. $d(O_{side} - N_{i+2}) < 3.5\,Å$
 iii. $∠(O_{side} - H - N_{i+2}) > 140\,°$
 iv. $d(O_{side} - N_{i+3}) < 3.5\,Å$
 v. $∠(O_{side} - H - N_{i+3}) > 140\,°$
 vi. $d(O_i - N_{i+3}) < 3.5\,Å$
 vii. $∠(O_i - H - N_{i+3}) > 140\,°$
 viii. $d(O_i - N_{i+4}) < 3.5\,Å$
 ix. $∠(O_i - H - N_{i+4}) > 140\,°$

 e. Class 3a (C3a)
 i. $N_i$ is Asp or Asn (ASX motif); Ser or Thr (ST motif)
 ii. $d(O_{side} - N_{i+3}) < 3.5\,Å$
 iii. $∠(O_{side} - H - N_{i+3}) > 140\,°$
 iv. $d(O_{side} - N_{i+2}) > 3.5\,Å$ or $∠(O_{side} - H - N_{i+2}) < 140\,°$
 v. $d(O_i - N_{i+3}) < 3.5\,Å$
 vi. $∠(O_i - H - N_{i+3}) > 140\,°$
 vii. $d(O_i - N_{i+4}) > 3.5\,Å$ or $∠(O_i - H - N_{i+4}) < 140\,°$

 f. Class 4 (C4)
 i. $N_i$ is Asp or Asn (ASX motif); Ser or Thr (ST motif)
 ii. $d(O_{side} - N_{i+2}) < 3.5\,Å$
 iii. $∠(O_{side} - H - N_{i+2}) > 140\,°$

 iv. $d(O_{side} - N_{i+3}) < 3.5$ Å

 v. $\angle(O_{side} - H - N_{i+3}) > 140$ °

 vi. $d(O_i - N_{i+4}) < 3.5$ Å

 vii. $\angle(O_i - H - N_{i+4}) > 140$ °

 viii. $d(O_i - N_{i+3}) > 3.5$ Å or $\angle(O_i - H - N_{i+3}) < 140$ °

 g. Class 4a (C4a)

 i. $N_i$ is Asp or Asn (ASX motif); Ser or Thr (ST motif)

 ii. $d(O_{side} - N_{i+3}) < 3.5$ Å

 iii. $\angle(O_{side} - H - N_{i+3}) > 140$ °

 iv. $d(O_{side} - N_{i+2}) > 3.5$ Å or $\angle(O_{side} - H - N_{i+2}) < 140$ °

 v. $d(O_i - N_{i+3}) < 3.5$ Å

 vi. $\angle(O_i - H - N_{i+3}) > 140$ °

 vii. $d(O_i - N_{i+4}) < 3.5$ Å

 viii. $\angle(O_i - H - N_{i+4}) > 140$ °

 h. Class 5 (C5)

 i. $N_i$ is Asp or Asn (ASX motif); Ser or Thr (ST motif)

 ii. $d(O_{side} - N_{i+3}) < 3.5$ Å

 iii. $\angle(O_{side} - H - N_{i+3}) > 140$ °

 iv. $d(O_{side} - N_{i+2}) > 3.5$ Å or $\angle(O_{side} - H - N_{i+2}) < 140$ °

 v. $d(O_i - N_{i+4}) < 3.5$ Å

 vi. $\angle(O_i - H - N_{i+4}) > 140$ °

 vii. $d(O_i - N_{i+3}) > 3.5$ Å or $\angle(O_i - H - N_{i+3}) < 140$ °

(iv) 68731 unique ASX motifs and 63718 unique ST motifs were found based on the criteria.

(v) Structural information including sequence, dihedral angles and distance were collected for these natural *N*-cap loops to build up the datasets.

## Search of hydrophobic patch and triangle in *N*-cap ASX motifs

A script was designed to search if the hydrophobic interactions between sidechains of (*N'*, *N4*) and (*N'*, *N3*) were likely to happen. The nearest carbon distance between sidechains of residue pairs was measured in the script, and if the distance was smaller than 4.5 Å, we assumed it was close enough to have hydrophobic interaction between the two sidechains. 2-element 1d arrays,(x,x), were used to describe the results. In this array, the first x represents interactions between sidechains of *N'* and *N4*, and the last for *N'* and *N3*. Presence of interactions between a sidechain pair will give a '1' for that element in the array, while absence gives '0'. For example, if a motif has interactions between (*N'*, *N4*) and (*N'*, *N3*), the result for it would be (1,1). Since *N3* and *N4* are naturally close in distance (*Cα⋯Cα* ~ 3.8 Å), the element (*N3*, *N4*) was regarded as 1 throughout, therefore, motifs with (1,1) are regarded as containing hydrophobic triangles, and motifs with (1,0) or (0,1) have hydrophobic patches.

## Syntheses of linear peptides

The linear peptides were synthesized using standard Fmoc peptide synthesis protocols (Supplementary Fig. 12) on TentaGel S RAM resin (capacity = 0.22 mmol/g) either on a LibertyBlue peptide synthesizer or manually. Couplings were carried out using 5 equivalents (equiv) of Fmoc-protected amino acid, 10 equiv of Oxyma, 10 equiv of DIC. Coupling reactions were allowed to proceed for 2 minutes in DMF at 90 °C under microwave, after which Fmoc deprotection was carried out with 20% (vol/vol) piperidine in DMF at 90 °C for 1 minutes. Upon completion of the peptide sequence assembly on resin and deprotection of the final Fmoc group, the *N*-terminal amine was acetylated by treatment with 25% (vol) acetic anhydride and 3.5% (vol) DIPEA in DMF. Cleavage and deprotection from the solid support was carried out using TFA/H$_2$O/TIPS (95/2.5/2.5 vol/vol) for 3 hours at room temperature (longer time are required if there is Arg on peptides). The resin was then filtered and washed with excess TFA. The isolated TFA solution was then evaporated, and the peptide was precipitated by adding cold diethyl ether. The peptide suspension in ether was centrifuged to pellet followed by decanting the ether. The pellet was then washed twice with cold diethyl ether. Purification of the resulting

peptides was achieved by high performance liquid chromatography (HPLC) on a reversed-phase C18 column to yield peptide with purity >95%. Purity and identity were assessed using ESI-MS and analytical HPLC on a C18 column.

## Synthesis of bicyclic capped peptides

The linear peptides were synthesized using standard Fmoc peptide synthesis protocols on TentaGel S RAM resin (capacity = 0.22 mmol/g) either on a LibertyBlue peptide synthesizer or manually. The three cystines were placed at *N'*, *N3* and *N4* at the *N*-terminus. Couplings were carried out using 5 equivalents (equiv) of Fmoc-protected amino acid, 10 equiv of Oxyma, 10 equiv of DIC. Coupling reactions were allowed to proceed for 2 min in DMF at 90 °C under microwave, after which Fmoc deprotection was carried out with 20% (vol/vol) piperidine in DMF at 90 °C for 1 min. Upon completion of the peptide sequence assembly on resin and deprotection of the final Fmoc group, cleavage and deprotection from the solid support was carried out using TFA/H$_2$O/TIPS (95/2.5/2.5 vol/vol) for 3 hours at room temperature (longer time is required for peptides with Arg). The resin was then filtered and washed with excess TFA. The isolated TFA solution was then evaporated, and the peptide was precipitated by adding cold diethyl ether. The peptide suspension in ether was centrifuged to pellet followed by decanting the ether. The pellet was then washed twice with cold diethyl ether and dissolved in 1:1 mixture of aqueous buffer (20 mM NH$_4$HCO$_3$) and ACN to make final concentration around 1 mM. 1.2 equiv of TBMB was added and the solution was stirred for 15 min at room temperature. Purification of the resulting peptides was achieved by high performance liquid chromatography (HPLC) on a reversed-phase C18 column to yield peptide with purity >95%. Purity and identity were assessed using ESI-MS and analytical HPLC on a C18 column.

## Circular dichroism (CD) experiments

Concentrations of the stock solution were determined by 288 nm absorption of Trp or 275 nm absorption of Tyr (and TMB, if present in peptides). Each sample was dissolved in 1x PBS (pH 7.4), 10% TFE/PBS or 20% TFE/PBS with the final concentration between 5 to 30 μM. CD spectra were acquired using circular dichroism spectrometer (Chirascan) equipped with a temperature controller using 1 mm cell at a scan speed of 0.5 nm/sec at indicated temperature. Each sample was scanned three times and the averaged spectrum was smoothed.

## Variable temperature CD experiment

Concentrations of the stock solution were determined by 288 nm absorption of Trp. Each sample was dissolved in 10% TFE/PBS. CD spectra were acquired using circular dichroism spectrometer (Chirascan) equipped with a temperature controller using 1 mm cell at a scan speed of 2 nm/sec at indicated temperature. Each sample was scanned three times and the averaged spectrum was smoothed. Temperature range (°C) between 5 to 90 with a step of 5 was used in the study. Mean residue ellipticity were calculated and used to make graphics.

## Extinction coefficient of tri-thiol alkylated TMB

Comparing UV spectra of BAMM *N*-capped peptides and linear peptides showed the thiol-alkylated TMB were likely to contribute to the absorbance significantly around 280 nm (Supplementary Fig 15a). Therefore, acquiring its extinction coefficient at 275 nm and 288 nm was necessary to accurately calculate peptide concentrations. A tri-thiol alkylated TMB derivative was made, and its UV spectrum was measured (Supplementary Fig 15b); merging UV spectra of equal-molar thiol-alkylated TMB with Tyr-peptides gave a very close spectrum as the BAMM *N*-capped peptide (Supplementary Fig 15c), proving the earlier assumption is correct. Significant UV contribution was observed at 275 nm (Tyr absorbance peak), but only negligible absorbance at 288 nm was found (Trp absorbance peak), suggesting it does not have significant effect for Trp-based peptides. Consequently, only

$\varepsilon$ at 275 nm was required. Based on Beer-Lambert law, we calculated its $\varepsilon$ (275 nm) = 555 $M^{-1}\cdot cm^{-1}$ (Supplementary Fig 15d). Therefore, the $\varepsilon$ (275 nm) for BAMM capped peptide is 1455 + 555 = 2010 $M^{-1}\cdot cm^{-1}$.

## Calculation of percent helicity

For simplicity, 'helicity' or 'helical' in the following text all refers to α-type. For each peptide, mean residue ellipticities ([θ]/(c*n), c = concentration of the peptide, n = number of amino acids in the peptide) were used to calculate % helicity, which is an empirical indicator of peptide helicity. Higher values of % helicity usually suggest the tested peptides are more helical, but does not necessarily mean what percentage of peptides are helical. For example, 100 % helicity does not mean the peptide is completely helical. Percent helicity was calculated based on the equation: [% helicity = $[θ]_{222}/[θ]_{max}$, where $[θ]_{max}$ = (-44000 + 250 $T$)(1 - $k/n$)] (n = number of amino acids in the peptide, $T$ = 25.0 °C). According to Baldwin's work[26,27], k refers to 'the number of non-H-bonded peptide mainchain carbonyls in a carboxyamidated peptide when it is completely helical', hence for linear peptide, k was 3, and for bicyclic peptides, depending on the number of residues extended after *N'* Cys, k could be 4 (no extension) or 5 (one residue extension, ie, *N''*), as illustrated in Supplementary Fig 16.

## MD analysis: bicyclic peptides with L-Cys and D-Cys

To find the best combination of L-Cys and D-Cys at (*N', N3, N4*), MD analysis using *Desmond* was executed. A linear ASX-capped 13-mer helical peptide with natural hydrophobic triangle (V, F, W) from PDB 1bvz was chosen as a control, and residues other than (V, F, W) and D were changed to A. Hydrophobic residues (V, F, W) were replaced by L- and D-Cys in the modified sequences, virtually cyclized with TBMB, and then minimized, which gave 8 bicyclic helical peptides (Ac-[C*DAAC*C*]$_{cyclo}$-AAAAAAA). Packages in *Schrödinger* was used in simulations. Each of these peptides were solvated by explicit water in *System Builder*, and then input to *Molecular Dynamics* for 50 ns and 1000 frames were kept during the simulation. Each frame was overlayed by backbone atoms with the initial helical structure. If the RMSD was smaller than 2 Å, the frame was considered helical. The number of helical frames was counted and converted to time (ns) to quantitated helical residence time during simulation. In the graphics, VFW represents linear control, C represents L-Cys while c for D-Cys.

## NMR experiments: 1D and 2D NMR spectroscopy

Spectra of tested peptides (sample prepared in 30 μL D$_2$O, 120 μL TFE-d$_3$ and 450 μL H$_2$O to make final concentration of 2–5 mM) were recorded on a Bruker AVANCE 500 with a cold probe at 33 °C. Water suppression was achieved by water gate pulse sequence with gradients for 1D $^1$H spectrum and by presaturation during relaxation delay for 2D $^1$H-$^1$H spectra. 2D $^1$H-$^1$H TOCSY spectrum was acquired with a mixing time of 80 ms, and ROESY spectrum with a mixing time of 200 ms. The spectra were acquired with 12 ppm spectra width and 2048 × 256 complex points. TOCSY spectrum was processed by TopSpin® to 2048 × 2048 complex points whereas ROESY spectrum to 2048 × 1024 complex points. $^3J_{NH-H\alpha}$ coupling constants were measured from 1D $^1$H spectra and 2D $^1$H-$^1$H TOCSY spectrum.

## Chemical shift index (CSI) calculation

CSI is a qualitative indicator of peptide/protein secondary structures[48]. By comparing α*H* chemical shifts with CSI residue-specific random coil α*H* shifts, the CSI for any peptide can be calculated. Consecutive α*H* upfield shifts (and most shifts >0.1 ppm) is a sign of α-helices, Consecutive α*H* downfield shifts (and most shifts <0.1 ppm) indicate β-strand. CSI of LDLL 12-mer and bicyclo 12-mer (Supplementary Fig. 29 and 36) indicates potential α-helical structures since D at *Ncap*. $\Delta\delta = \delta_{\alpha H}$ (experimental) - $\delta_{\alpha H}$ (random).

## Elucidation of solution conformations by NMR constraints

ROEs in the ROESY spectrum were assigned to generate distance restraints by equation $v = k/(u^6)$ where v is cross peak intensity, u is distance between two hydrogens, and k can be chosen by inputting known NOEs of certain distances into the equation[49]. $^3J_{NH-H\alpha}$ coupling constants were measured from 1D $^1$H spectra and 2D $^1$H-$^1$H TOCSY spectrum to generate dihedral constraints by {$J(θ)$ = 6.98 * (cos(θ))$^2$ − (1.38*cos(θ)) + 1.72, where $θ = |Φ-60|$}[49]. Distance restraints (summarized in Supplementary Tables 2, 5) and dihedral constraints (in Supplementary Table 3, 6) were then utilized for structure calculation using Macromodel package in *Schrödinger*. An optimized conformational sampling method, Enhanced MCMM[36], was used to exhaust possible conformations of the input structure and those unmatched conformations were filtered out by the constraints. Conformers within 3 kcal/mol (12.6 kJ/mol) to the lowest-energy conformer were then interpreted as the solution structural ensemble.

## Amide H-D exchange study

Lyophilized samples of peptides LDLL 12-mer and bicyclo 12-mer from the above experiments were dissolved in 600 μL of D2O/ TFE-d$_3$ mixture (4:1) to initialize the *H-D* exchange. The pH of the solution was confirmed. Spectra were recorded on a preshimmed Bruker AVANCE 500 MHz spectrometer. The recorded temperature was 33 °C. The intensity changes for each amide proton were determined by monitoring either the *NH* peaks on 1D spectra or the cross-peaks between *NH* and *RH* on 2D TOCSY spectra when overlapping was severe. The peak intensity data was fit into one phase exponential equation to get the exchange rate constants using GraphPad Prism 6.0 program. The rate constants were input into Englander lab's spreadsheet (https://hx2.med.upenn.edu/download.html) to calculate protection factor and stabilization energy[37].

## In vitro peptide stability in human serum[50]

Male human serum (H-4522) was obtained from Sigma Aldrich (Missouri, USA). The tested peptide was mixed to make a final peptide concentration of 150 μM in 25% human serum diluted in RPMI medium. Before the addition of the tested peptide, the diluted serum solution was temperature-equilibrated at 37 ± 1 °C for 15 min. The initial time is recorded as peptides were added. The mixture was incubated at 37 °C and an aliquot of 40 μL was removed at different time intervals. The aliquot was mixed with 160 μL cold methanol and incubated at 4 °C for 15 min before centrifuging for 10 min to precipitate serum protein. The supernatant was analyzed in analytical RP-HPLC to calculate the amount of tested peptides, and in LCMS to analyze degradation products. The eluted peptides were detected by absorbance at 210 nm and quantified by their peak areas relative to the initial peak areas (0 h).

## Direct FP assay to measure K$_d$ of fluorescent peptides

Polarized fluorescence intensities were measured on Synergy H4 plate reader with excitation and emission wavelengths of 485 and 535 nm with 20 nm band widths, respectively. The optics position is Top 510 nm with Gain set as 100. Read Speed is normal and read height is 7 mm. Cyclin proteins were bought from Sino Biological and after reconstitution it should be used immediately. Leaving proteins in freezer for weeks could yield either precipitations or significantly increased background fluorescence. To each well, 10 μL 40 nM of fluorescent peptides in PBS was added. Various concentrations of cyclin E (from Sino Biological, Inc.) were prepared (0.16–4.8 μM) and 10 μL were added to each well. The plate was shaken in the dark for 3 h. Wells containing only the peptide and PBS buffer were used as a control. Each mixture was prepared in duplicates. K$_d$ values of peptides were fitted by the equations below:

- $P = f_{bound} * P_{bound} + f_{free} * P_{free}$
- $f_{bound} = \frac{a - \sqrt{a^2 - 4 * R_{total} * L_{total}}}{2 * L_{total}}$, $a = (K_d + R_{total} + L_{total})$
- $f_{bound} + f_{free} = 1$

- $P = \frac{(I_{II} - I_{I})}{(I_{II} + I_{I})} = Y, R_{total} = X$
- $L_{total}$ is the amount of tracer, a constant
- $P_{bound}, P_{free}$ and $K_d$ are left for fitting in Graphpad Prism

## Trp fluorescence quenching assay

Fluorescence quenching of the single tryptophan in cyclin E was measured by using a Cary Eclipse (Varian) equipped with a front-face fluorescence accessory at 25 °C, by using 10-nm excitation and 10 nm emission bandwidths. The excitation wavelength was 295 nm and the emission spectra were measured between 315 and 395 nm. Titrations were performed in a 0.7-ml quartz fluorescence cuvette containing 0.5 mL 0.8 µM protein in PBS buffer, pH 7.4, and by the successive addition of 0.5–1 µL of compound stock solutions. After each addition, the cuvette was shaken to enable fast mixture, and there were 10 mins interval between two additions to ensure equilibrium of the mixture. The measurements were repeated at least three times at each concentration. Fluorescence intensities at 338 nm at increasing concentrations of the quencher were collected. Dissociation equilibrium constant ($K_d$) values were determined by fitting data to the equation below[42,43]. The fitting plot was reformatted to make Y equals $f_{bound}$ in Fig. 6g to easily visualize the binding processes.

- $F_{obs} = F_{free} * f_{free} + F_{bound} * f_{bound}$
- $f_{free} + f_{bound} = 1$
- $f_{bound} = \frac{a}{c_{protein}}$

- $a = \frac{(-b - \sqrt{b^2 - 4*d})}{2}$
- $b = -(K_d + c_{ligand} + c_{protein})$
- $d = c_{ligand} * c_{protein}$
- $F_{obs} = Y, c_{ligand} = X$
- $c_{protein}$ is total concentration of protein, a constant
- $F_{free}, F_{bound}$ and $K_d$ are left for fitting in Graphpad Prism

## Characterization of purified peptides

Purity of peptides were measured in analytical HPLC runs using a Zorbax SB-C18 column (Agilent) with a 20 minute gradient between {5% solvent A (99.9% water, 0.1% TFA), 95% solvent B (99.9% acetonitrile, 0.1% TFA)}, and {95% solvent A, 5% solvent B}. Absorption traces at 210 nm were presented along with the retention time for different peptides. Expected masses of peptides were calculated from ChemDraw, and observed masses are M + H⁺ peaks from ESI-MS spectra.

## Reporting summary

Further information on research design is available in the Nature Portfolio Reporting Summary linked to this article.

## Data availability

Data supporting the findings of this study are available within the paper and its Supplementary Information. Datasets generated in this study have been uploaded in Supplementary Information. The structural data that support the findings in this study have been deposited in the Protein Data Bank with the coordinate accession numbers 8UN8 and 8UTX. Raw data of main text Figs are provided with this paper. Other data that support the findings of this study are available upon request to the corresponding author. Source data are provided with this paper.

## Code availability

Codes used in this manuscript was written to collect unique ASX motifs from PDB and then statistically analyze them to find trends. All are available online at https://github.com/burgess-lab-tamu/ASX-ST-Search. They are also deposited in Zenodo with https://doi.org/10.5281/zenodo.10819710.

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

## Acknowledgements

Financial support was provided by NIH RO1EY029645, NIH R21NS130471-01A1, and the Texas A&M University T3-Grants Program (246292 – 00000) awarded to K.B.

## Author contributions

TM developed the algorithm for ASX/ST Search, collected data from it, made correlations, planned and performed most of the experimental work with input from KB; TM prepared the SI material. DN assisted with some syntheses of the natural peptides and the BAMM peptidomimetics, and some of the CD studies. KB conceived the overall direction of the work, and wrote the manuscript with comments from TM, DN and ZG. KB and TM together composed and refined the graphics.

## Competing interests

The authors declare no competing interests.
