## [Peer Review File · Nature Communications]

Bioinformatics Leading To Conveniently Accessible, Helix Enforcing, Bicyclic ASX Motif Mimics (BAMMs)Editorial Note: This manuscript has been previously reviewed at another journal that is not operating a transparent peer review scheme. This document only contains reviewer comments and rebuttal letters for versions considered at *Nature Communications*.

REVIEWER COMMENTS

Reviewer #3 (Remarks to the Author):

With the helix induction in polypeptides of varying lengths, the authors demonstrate the robustness of their approach to target biologically relevant protein-protein interactions. However, I have strong reservations about the cell-permeability study, which is only done through FACS. No microscopy was done to determine whether the peptides are within the cell or adhered to the cell surface, purely due to the amplified hydrophobicity arising from the cyclized motif. Furthermore, no detailed studies have been done to elucidate the mechanism of peptide uptake. Moreover, I could not find the concentrations of the peptides used in the permeability studies. No dose-dependent data is presented.

Perhaps limiting the study to the helix induction and its utility in targeting PPIs *in vitro* would have been optimal.

Nonetheless, while I was in support of the study while it was being reviewed elsewhere, at present, I am not in support of the publication of the work in *Nature Communications* due to the reasons stated below:

The study's uniqueness lies in identifying the hydrophobic triangle at the N-term of the helices, which acts to cap the helix and provide structural stability. The authors cleverly used this information to incorporate cysteine residues strategically into model peptides and form a bicyclic motif using 1,3,5-tribromomethylbenzene to induce helicity. However, to my surprise, the authors reported an identical strategy to induce helicity in polypeptides, where a hydrophobic triangle is formed in polypeptides at the C-terminal, earlier this year in *ACS Central Science* (ref. no. 48 in the present manuscript).

What is also surprising is that the authors mined the identical data set from the Protein Data Bank (Fig. 2a in either work) to derive the information about the presence of hydrophobic triangle either at the N-terminus (present manuscript) or at the C-terminus (*ACS Central Sci.*)

in polypeptides. Therefore, several controls, descriptions, and graphics are comparable in either work. In my opinion, this significantly compromises the novelty of the work for publication in Nature Communications.

TITLE: Conveniently Accessible, Bicyclic ASX Motif Mimics (BAMMs) Inspired By Bioinformatics

We thank reviewer 3's comments, which help us significantly improve this manuscript. Our answers to questions from reviewer 3 are as outlined below. Our responses to reviewer 1's comments from the 1st-round review are attached after this.

Reviewer 3

1. However, I have strong reservations about the cell-permeability study, which is only done through FACS. No microscopy was done to determine whether the peptides are within the cell or adhered to the cell surface, purely due to the amplified hydrophobicity arising from the cyclized motif. Furthermore, no detailed studies have been done to elucidate the mechanism of peptide uptake. Moreover, I could not find the concentrations of the peptides used in the permeability studies. No dose-dependent data is presented.

Perhaps limiting the study to the helix induction and its utility in targeting PPIs in vitro would have been optimal.

We agree with Reviewer 3's comment "limiting the study to the helix induction and its utility in targeting PPIs", so we decided to simply take this data out, see above. Mechanism of uptake is a different issue than seeing uptake. Determination of mechanism requires many experiments and even then is rarely definitive. This manuscript does not need cell uptake experiments to stand alone.

2. However, to my surprise, the authors reported an identical strategy to induce helicity in polypeptides, where a hydrophobic triangle is formed in polypeptides at the C-terminal, earlier this year in ACS Central Science (ref. no. 48 in the present manuscript).

This manuscript described this work and cited the reference last time it was reviewed by Referee 3, so this should not be a surprise.

True, BSMs and BAMMs were both discovered based on findings from bioinformatics, but the datamining approaches were fundamentally different, and the design patterns are not the same. These are the differences:

- (i) Bioinformatics leading to BSMs were based on Schellman loop in proteins, but BAMMs are based on mining ASX motifs. These are two different protein motifs, so their data mining processes are separate, and independent.
- (ii) After we discovered the hydrophobic triangle in Schellman loops there was no evidence that a similar triangle would be found in ASX motifs, we discovered it for this manuscript.
- (iii) Dispositions of the hydrophobic residues in ASX structures are *different* to those in Schellman loops. Thus, after the BSM work we did not know hydrophobic triangles even existed at the *N*-terminus, and could not possibly have known that the positions and shapes of the triangles were different; we had to work this out. Very briefly (but see new text for full explanation), the patterns are (Ψ is hydrophobic, X is any amino acid, and the cap residue is underlined):

C-terminus: Ψ -X-X- Ψ -G- Ψ and in BSMs helix-C-X-X-C-G-C

N-terminus: Ψ -D-X-X- Ψ - Ψ and in BSMs C-D-X-X-C-C-helix

The caps are in different positions so three residues of BSMs are integrated into the helix, whereas BAMMs have four. We discovered these fundamental differences then capitalized on them. A person trying to simply reproduce the BSM shape and pattern at the *N*-terminus would have failed to obtain helical induction.

The discussions above indicate the two triangles, despite with similar names (triangle), are indeed two different structures. This difference also reflects in their bicyclic mimics: Bamm and BSM. In this revision, we compared helix-enforcing mechanisms of the two capping systems, and concluded that Bamm is more helix-inducing, while BSM is more helix-stabilizing, hence they have complementary functions. This shows the difference of the two caps which mimic *N*- and *C*-hydrophobic triangles, respectively.

3. What is also surprising is that the authors mined the identical data set from the Protein Data Bank (Fig. 2a in either work) to derive the information about the presence of hydrophobic triangle either at the *N*-terminus (present manuscript) or at the *C*-terminus (ACS Central Sci.) in polypeptides.

I think the origin of this misunderstanding is clear from our response 2 above, *ie*:

- (i) The two studies feature different dataset: this paper is based on ASX motifs, but ref 48 features Schellman loops. Independent data mining processes were done to obtain the two distinct datasets.
- (ii) We found the *N*-triangles in Asx motifs, and *C*-triangles in Schellman loops. These two are indeed two different structures. It is impossible to discover the pattern of *N*-triangles by directly mimicking the residue dispositions in *C*-triangles without careful bioinformatic studies (see response 2 above and new Fig 7).
- (iii) The difference in *N*- and *C*-triangles also is reflected in their bicyclic cap mimics, as described in responses 2 above and Fig 7 with surrounding text.

4. Therefore, several controls, descriptions, and graphics are comparable in either work.

Only the same linear peptides in model Ala-rich sequences and one of the five biological systems were used in the two papers, no other compounds feature in both. In addition, this work features more validation on real peptides, a detailed NMR study, serum stability assays, and the Bamm BSM comparison; none of these features have correspondence in the BSMs paper.

True CD studies and solution conformations by NMR experiments, were used in both works, but these techniques are common to many papers on peptides.

REVIEWER COMMENTS

Reviewer #1 (Remarks to the Author):

General Comments.

The idea of hydrophobic triangles is interesting and the experimental execution simple. My comments below add to and agree with comments in previous reviews of this paper. The authors have improved some key technical aspects of this paper from an earlier version and responded to many of the experimental points raised from a previous review for another journal. In this revision, the authors in particular have now included (1) model peptide sequences, (2) used less of the helix-favoring TFE solvent for CD measurements to report on structure (10% instead of 20%), (3) recorded some helix-defining i to $i+4$ ROEs for comparison with i to $i+2$ ROEs, (4) included Ramachandran analyses of phi and psi angles. This extra information gives more confidence for claims of helicity for some of the peptides or at least regions within them.

The message in this paper is conceptually simple. However, the experimental evidence is presented in the main text in a complex and semi-scientific way. The main figures and main text are mostly written as abbreviated summaries of work rather than reporting original data. This makes figures/tables in the main text hard to assess and understand - the reader is required to accept these summaries on face value. Legends for these figures/tables lack almost all of the essential detail and information required to understand the summaries and what they refer to. One has to spend 90% of the reading time on the SI component of the manuscript and try to sift through the detail of the large SI section to dissect out any findings and draw conclusions. This is not easy to do, even for workers in this field. For these reasons, I am not persuaded that this manuscript is particularly suited for a general scientific audience. I think general readers may be confused/bamboozled by the substance and evidence for conclusions drawn (or accept the conclusions without scrutinizing the data).

Some specific comments:

It would be helpful to number the pages in the SI section.

Title.

The title has two acronyms which is not normal. I would support keeping the ASX acronym, which is searchable and has meaning for researchers in this field. I don't think BAMMs is needed in the title and will be defined for the first time later in the paper.

Introduction.

p.3 L3 "interface α -helix mimicry (abbreviated to helical mimicry in this paper)"

An alpha helix is not the only kind of helix. If this abbreviation is to be made, then it follows that the helices to be described must have specific dihedral angles (ϕ , ψ) consistent with a right-handed alpha helix (and not be a distorted helix). Helical mimicry may be a better term rather than being defined as "alpha" helical mimicry.

p.3 L8 "but there is no evidence stapling is uniformly more effective"

This is not really correct. An appropriate "alpha" helix-inducing staple in a peptide induces helicity in both directions whereas a cap (N or C terminal) can only induce helicity in one direction and can use only half as many H-bonding atoms to do this. This is important. Comparative studies using the same staple at the N- vs C- terminus vs in the middle have shown more effective helix induction from the middle, and better helix induction from the C- than the N-terminus (e.g. ACIE 2016, 55, 8275). The problem seems to be more that different staples have wildly different capacities to stabilize a true "alpha" helix and frequently also involve structural contributions from 3-10 helix, beta or other turn motifs, and other variants.

p.3 Last paragraph. I think the reason that capping is used less frequently is because it only works in one direction and the influence of the cap falls away with each successive turn, leading (as stated) to the other end fraying and being susceptible to proteolysis. The Arora approach (and earlier iterations on which it was based) worked quite well but, as stated, is synthetically inconvenient. The Li approach involved conformationally flexible monocycles. The Fairlie approach used synthetically convenient monocycles that were conformational rigid but this may perhaps limit fitting to some proteins.

Results.

p.6-8 Bioinformatics: I am not able to comment constructively on the veracity, feasibility or effectiveness of this approach as merely outlined/overviewed in Figure 2. There is no original data in this figure. One needs to understand the detail of the extensive SI section (S1-S11) to interpret this. I found this section difficult to assess and draw conclusions without repeating the work so I cannot comment on its veracity. Personally, I think this part should be published separately and scrutinized by a bioinformatician rather than data dumped in the SI.

p.9 Peptide Syntheses (sections C and M, SI): The analytical HPLCs look to be of pure peptides with the required MS data presented. Thus the identities of the peptides is demonstrated.

Figure 3 is a graphical summary rather than original data - one needs to study the SI to reach any conclusions.

CD and NMR Spectra:

A positive component of this revised manuscript is that CD spectra are now re-recorded for peptides in 10% TFE in this version (previously in 20% TFE which promotes helicity). CD spectra can provide a quick summary of structure, although they reflect all contributing ensemble structures in the mix and slight changes in the component structural mix make it difficult to compare, for example, the amount of alpha helicity. The strong CD band at 223 nm is claimed to be due to "curvature effect" for Aline rich peptides and an early reference is given for this. I could show 20 references that show coupled beta strands/sheets of different complexity also generate a dominant peak at 220-230 nm without the 208 nm band. There are many examples of "alpha helix" looking CD spectra that are not alpha helices in NMR structures. The CD spectra in Figs S15 and S16 (and S18-20) show some examples where the structure is unlikely to be a pure "alpha" helix - there is both a shift of the 222 nM signal to longer wavelengths and changes to the ratio of the 222/208 peaks. For this reason, the use of "% helicity" at 222nm to compare "alpha" helix induction in Figure 3 main text can be misleading, in my opinion. Similarly the discussion in the SI of "improving

helicity by 7%" is misleading. These quantitative % comparisons are only really helpful for the largest changes where a qualitative structural change is clear.

Both the ratio and shape of the bands at 204? and 223/224 nm in Fig 4d for the bicycle 12mer W-[C2-D-A1-A4-C3-C1]cyclo-A2-A5-A3-K-A6-NH₂ are not ideal for a CD spectrum of an "alpha" helical peptide. Its NMR structure and Ramachandran plot do however support helix-like phi and psi angles for the segment between A1 to K, but not for W-C2 or A which may explain distortion in the CD spectra from an "alpha helical" peptide. The authors also have now recorded CD spectra on variable concentrations of peptide (as requested) and concluded that these CD spectra are independent of concentration (Fig S21), suggesting little aggregation. This is possible, but all of their CD spectra only show the region above a wavelength of 200 nm. Full CD spectra for peptides are usually shown from 185 to 260 nm since an "alpha helix" CD shows a dominant third definitive strong positive molar ellipticity around 195 nm. Bands lower than 200 nm are much more sensitive to impurity and concentration of solutes and also usually better evidence of concentration independence of structure - I presume the aromatic ring of the cap absorbs in this region and that this is why the authors do not show any CD bands below 200 nm.

NMR spectra for the above peptide are mostly of very good quality. There are three detectable i to i+4 ROEs supporting a helical segment and few i to i+2 ROEs are shown (consistent with an alpha helical segment). Consecutive amide coupling constants below 6 Hz for the segment A4-C3-C1]cyclo-A2-A5-A3-K are in agreement with the i to i+4 ROEs, supporting helicity for this part of the peptide (Fig S34, Fig S36). I found no errors in their CD and NMR methods. I think the authors should therefore draw the conclusion that part of the peptide is helical.

Manuscript ID: NCOMMS-23-37724A

TITLE: Bioinformatics Leading To Conveniently Accessible, Helix Enforcing, Bicyclic ASX Motif Mimics (BAMMs)

Below is a comprehensive list of the referees' suggestions for changes in this font, and our corresponding actions or responses.

Reviewer 1

1. the experimental evidence is presented in the main text in a complex and semi-scientific way. The main figures and main text are mostly written as abbreviated summaries of work rather than reporting original data. This makes figures/tables in the main text hard to assess and understand.

.....

Legends for these figures/tables lack almost all of the essential detail and information required to understand the summaries and what they refer to. One has to spend 90% of the reading time on the SI component of the manuscript and try to sift through the detail of the large SI section to dissect out any findings and draw conclusions.

We remade parts of main text Fig 2 and 3 by incorporating more original computational and experimental data, expanded and modified the main text describing them by moving relevant details from the SI to the main text. Please read the yellow-marked paragraphs to check changes.

We also expanded Fig captions to include more details.

2. It would be helpful to number the pages in the SI section..

Page numbers added.

3. The title has two acronyms which is not normal. I would support keeping the ASX acronym, which is searchable and has meaning for researchers in this field. I don't think BAMMs is needed in the title and will be defined for the first time later in the paper.

We respectfully disagree with this point. Bamm is defined in the title, and every element of that definition is a well-known term so the abbreviation does not cause confusion. Further, it is probable people will search for "Bamm" in future and it will be easier to find in the title than in the text.

4. p3. L3 "interface α -helix mimicry (abbreviated to helical mimicry in this paper)"
An alpha helix is not the only kind of helix. If this abbreviation is to be made, then it follows that the helices to be described must have specific dihedral angles (ϕ , ψ) consistent with a right-handed alpha helix (and not be a distorted helix). Helical mimicry may be a better term rather than being defined as "alpha" helical mimicry.

We removed ' α -' as suggested.

5. p.3 L8 "but there is no evidence stapling is uniformly more effective". This is not really correct. An appropriate "alpha" helix-inducing staple in a peptide induces helicity in both directions whereas a cap (N or C terminal) can only induce helicity in one direction and can use only half as many H-bonding atoms to do this. This is important. Comparative studies using the same staple at the N- vs C- terminus vs in the middle have shown more effective helix induction from the middle, and better helix induction from the C- than the N-terminus (e.g. ACIE 2016, 55, 8275). The problem seems to be more that different staples have wildly different capacities to stabilize a true "alpha" helix and frequently also involve structural contributions from 3-10 helix, beta or other turn motifs, and other variants.

We removed that phrase, and added another to catch the referees point about the composition of the staple influencing degree of α -helicity. Thus the paragraph now reads

"Most stapled peptides feature exocyclic *N*- and *C*-terminal peptide regions vulnerable to conformational fraying, and proteolytic trimming by exopeptidases. Further, compositions of staples used varies widely, and this can impact populations of true α -helical conformations induced. Staples have been studied more extensively than the alternative: synthetic caps.^{5,6}

Capping motifs are different to staples. They kink terminal backbones out of helicity to satisfy overhanging *H*-bond donors or acceptors. *N*-Caps may flex but cannot fray, hence they protect *N*-termini from proteolytic degradation. Those properties provide advantages, but the current methodologies for installing *N*-caps have limitations."

6. p.3 Last paragraph. I think the reason that capping is used less frequently is because it only works in one direction and the influence of the cap falls away with each successive turn, leading (as stated) to the other end fraying and being susceptible to proteolysis. The Arora approach (and earlier iterations on which it was based) worked quite well but, as stated, is synthetically inconvenient. The Li approach involved conformationally flexible monocycles. The Fairlie approach used synthetically convenient monocycles that were conformational rigid but this may perhaps limit fitting to some proteins.

To avoid misunderstanding, we modified the first sentence of this paragraph to

" Current *N*-cap systems are inconvenient to prepare and/or give relatively flexible cap mimics."

The following point is incidental since the referee did not request a change to discuss the Fairlie work

We are not convinced Fairlie's are 'caps'. Cap should terminate helices at one end and induce/continue helicity at the other end. His [KAXAD]_{cyclo} is a staple to stabilize and induce helicity in **two** directions at the middle of a sequence, for example, Chembiochem 2017, 18, 2087 and many lactam stapled peptides summarized in the review Chem Soc Rev 2015, 44, 91. Fairlie's ACIE 2016, 55, 8275 had 'KAXAD' at both termini (peptides 6 – 9 in this ref), all residues, including the terminal 'KAXAD', fell in α -helical space in Ramachandran plots, suggesting 'KAXAD' is a 'staple' and the termini could be extended to continue the α -helical conformations.

7. p.6-8 Bioinformatics: I am not able to comment constructively on the veracity, feasibility or effectiveness of this approach as merely outlined/overviewed in Figure 2. There is no original data in this figure. One needs to understand the detail of the extensive SI section (S1-S11) to interpret this. I found this section difficult to assess and draw conclusions without repeating the work so I cannot comment on its veracity. Personally, I think this part should be published separately and scrutinized by a bioinformatician rather than data dumped in the SI.

[handling editor's name redacted] please recall you suggested we leave the bioinformatics in. However, we have made some changes in response to the comments above.

Fig 2 describes how we discovered *N*-terminal hydrophobic triangles from the bioinformatic process, and which residues (*N'*, *N3* and *N4*) are most commonly found in those triangles. The reviewer suggests adding more original data into the Fig. We made following changes even though readers wishing to repeat, rather than just understand, the work would naturally go to the SI.

1) We made a new Fig 2c-e by combining SI Fig S7 and original Fig 2e to describe details of how we discovered the triangles. Absolute numbers of ASX motifs in the subset, potential patches and triangles are given in new Fig 2d.

2) Text describing Fig 2 includes more details (from the SI) as the reviewer requested. The Fig 2 caption has been modified to include more details too.

However, we respectfully disagree with several points from the reviewer on Fig 2

'There is no original data in this figure.'

Original data for the bioinformatic work are Python codes and enormous mined datasets. It is not realistic to give the codes or datasets in the main text. Instead, we uploaded the codes in open-source GitHub and the mined datasets in supplementary materials for review or other purposes. Readers/reviewers are free to repeat the work using those codes and datasets with a regular PC. In this revision we provide key technical values in the Fig 2 and describing text.

One needs to understand the detail of the extensive SI section (S1-S11) to interpret this

Not all S1-S11 are related to Fig 2. Fig 2 is to describe discovery of *N*-terminal hydrophobic triangles, and favorable residues at the triangular positions, so we only show relevant data to illustrate that. We think Fig 2 now is self-explanatory and provides a clear description.

Personally, I think this part should be published separately and scrutinized by a bioinformatician rather than data dumped in the SI.

N-terminal hydrophobic triangle was discovered from this simple and straightforward bioinformatic process. The manuscript follows a clear, logical, sequence: discovery of a new motif – motif mimicry – applications in different peptide sequences.

8. Figure 3 is a graphical summary rather than original data - one needs to study the SI to reach any conclusions.

We have remade Fig 3 with experimental CD spectra (Fig 3b-d), and moved the SI details describing the CD spectra to the main text for the new Fig 3.

9. The strong CD band at 223 nm is claimed to be due to "curvature effect" for Ala-rich peptides and an early reference is given for this. I could show 20 references that show coupled beta strands/sheets of different complexity also generate a dominant peak at 220-230 nm without the 208 nm band. There are many examples of "alpha helix" looking CD spectra that are not alpha helices in NMR structures..

We do not agree the strong CD peak at 223 nm of bicyclo 12-mer may be related to beta-structure, because the conclusion that it is due to the curvature effect is supported by other experiments: (i) CD on different BMM capped sequences; (ii) variable-temperature CD experiments of bicyclo 12-mer; and, (iii) NMR results

In more detail, we tested six BMM capped peptide, one Ala-rich and the other five with natural sequences/not Ala-rich. Only the Ala-rich one shows the increased 222/208 ratios in its CD spectra while the other five showed more standard α -helical shape (222/208 ratios \sim 0.8 -1.0). This implies increased ratios in bicyclo 12-mer were more likely due to its Ala-rich sequence.

Ala-rich sequences are more helix-inducing. The reference we cited showed once *N*-capped, Ala-rich peptides reach maximal helicity, further decreasing temperature or increasing the ratios of organic solvent would increase 222/208 ratios of tested peptides in CD spectra; more specifically, 208 ellipticity is unchanged, 222 ellipticity is increasing, making 222/208 ratios increase. This is exactly what we observed in the VTCD experiments for bicyclo 12-mer from \sim 35 – 5 °C (208 peak unchanged but 222 peaks became more negative, making 222/208 ratios increase from 1.0 to 1.25 from 35 to 5 °C). This is an evidence of curvature effect observed in the *N*-capped, Ala-rich peptides (see the ref).

Further, Our NMR and calculations also show bicyclo 12-mer have a major conformation: a bicyclic *N*-cap followed by helical residues starting at N1. The reviewer is satisfied with our NMR methods and interpretations.

10. The CD spectra in Figs S15 and S16 (and S18-20) show some examples where the structure is unlikely to be a pure "alpha" helix - there is both a shift of the 222 nM signal to longer wavelengths and changes to the ratio of the 222/208 peaks. For this reason, the use of "% helicity" at 222nm to compare "alpha" helix induction in Figure 3 main text can be misleading, in my opinion. Similarly the discussion in the SI of "improving helicity by 7%" is misleading. These quantitative % comparisons are only really helpful for the largest changes where a qualitative structural change is clear.

We agree the linear sequence **LDLF** which has larger 222/208 ratios, and shifts its peak to \sim 226 nm in its CD spectrum may not be a pure 'alpha'-helix, and we explicitly point this out in the caption for Fig S15.

We removed sentences in main text and SI saying "improving helicity by 7%".

11. Its NMR structure and Ramachandran plot do however support helix-like phi and psi angles for the segment between A1 to K, but not for W-C2 or A which may explain distortion in the CD spectra from an "alpha helical" peptide.

W-C2 are *N''* and *N'*, ie before *Ncap* (which is D), so expected to be non-helical (see Fig 7a), and NMR showed they are indeed non-helical. The last **A** at C-terminus (WCDAACCAAK**A**) is commonly effected by solvents and probably too far from the bicyclic *Ncap* to be constrained in helical conformations. These are characteristics of capped helix; some of the capping residues are not, and should not be, helical.

12. All of their CD spectra only show the region above a wavelength of 200 nm. Full CD spectra for peptides are usually shown from 185 to 260 nm since an "alpha helix" CD shows a dominant third definitive strong positive molar ellipticity around 195 nm. Bands lower than 200 nm are much more sensitive to impurity and concentration of solutes and also usually better evidence of concentration independence of structure - I presume the aromatic ring of the cap absorbs in this region and that this is why the authors do not show any CD bands below 200 nm.

We did want to record 185-200 nm region into our CD spectra, but due to the instrument limitations, the CD curves before 200 nm were noisy, and difficult to smooth. We were unable to include this band into our presentations.

13. Consecutive amide coupling constants below 6 Hz for the segment A4-C3-C1]cyclo-A2-A5-A3-K are in agreement with the i to $i+4$ ROEs, supporting helicity for this part of the peptide (Fig S34, Fig S36). I found no errors in their CD and NMR methods. I think the authors should therefore draw the conclusion that part of the peptide is helical.

Our BMM *N-cap* was designed to stabilize and induce helicity for residues after *Ncap*, ie *N1* and afterwards; residues before *N1* are *N"-N'-Ncap* which are used to form the cap hence naturally non-helical (see answer 11). To be clear, in the bicyclo 12-mer, *Ncap* is Asp and *N1* is A1, so residues beyond *N1*, W-C2-D, are expected to be non-helical. This is in agreement with our NMR measurements. We also expected BMM would enforce residues after *Ncap* (D in this case) to be helical, that is A1-A4-C3-C1-A2-A5-A3-K-A6. For those expected helical residues, A6 deviated slightly since its position at the C-terminus is distal from the helix-inducing *N-cap*.

The paragraph above means our BMM helical mimics would always include a few non-helical residues before *N1* to form the cap, so they were never expected to be completely α -helical from the starting residue in the sequence. We added a paragraph to make this point very clear in text describing the NMR structures.